# Differential Privacy for Growing Databases

**Rachel Cummings**[*]
Georgia Institute of Technology
rachelc@gatech.edu

**Sara Krehbiel**[*]
University of Richmond
krehbiel@richmond.edu

**Kevin A. Lai**[*]
Georgia Institute of Technology
kevinlai@gatech.edu

**Uthaipon Tantipongpipat**[*]
Georgia Institute of Technology
tao@gatech.edu

## Abstract

The large majority of differentially private algorithms focus on the *static setting*, where queries are made on an unchanging database. This is unsuitable for the myriad applications involving databases that grow over time. To address this gap in the literature, we consider the *dynamic setting*, in which new data arrive over time. Previous results in this setting have been limited to answering a single non-adaptive query repeatedly as the database grows [DNPR10, CSS11]. In contrast, we provide tools for richer and more adaptive analysis of growing databases. Our first contribution is a novel modification of the private multiplicative weights algorithm of [HR10], which provides accurate analysis of exponentially many adaptive linear queries (an expressive query class including all counting queries) for a static database. Our modification maintains the accuracy guarantee of the static setting even as the database grows without bound. Our second contribution is a set of general results which show that many other private and accurate algorithms can be immediately extended to the dynamic setting by rerunning them at appropriate points of data growth with minimal loss of accuracy, even when data growth is unbounded.

## 1 Introduction

Differential privacy is a well-studied framework for data privacy. First defined by [DMNS06], differential privacy gives a mathematically rigorous worst-case bound on the maximum amount of information that can be learned about any one individual's data from the output of an algorithm. The theoretical computer science community has been prolific in designing differentially private algorithms that provide accuracy guarantees for a wide variety of machine learning problems (see [JLE14] for a survey). Differentially private algorithms have also begun to be implemented in practice by major organizations such as Apple, Google, Uber, and the United Status Census Bureau.

The large majority of work in differential privacy focuses on the *static setting*, in which adaptive or non-adaptive queries are made on an unchanging database. However, this is unsuitable for the myriad applications involving databases that grow over time. For example, a hospital may want to publish updated statistics on its growing database of patients, or a company may want to maintain an up-to-date classifier for its expanding user base. To harness the value of growing databases and keep up with data analysis needs, guarantees of private machine learning algorithms and other statistical tools must apply not just to fixed databases but also to dynamic databases.

To address this gap in the literature, we consider the *dynamic setting*, in which new data arrive over time. Previous results in this setting have been limited to answering a single non-adaptive query

---

[*]Author order is alphabetical and all authors contributed equally.

repeatedly as the database grows [DNPR10, CSS11]. In contrast, we provide tools for richer and more adaptive analysis of growing databases. Our first contribution is a novel modification of the private multiplicative weights algorithm of [HR10], which provides accurate analysis of exponentially many adaptive linear queries (an expressive query class including all counting queries) for a static database. Our modification maintains the accuracy guarantee of the static setting even in the presence of unbounded data growth. Our second contribution is a set of more general techniques to adapt any existing algorithm providing privacy and accuracy in the static setting to the dynamic setting. Our techniques schedule black box access to a static algorithm as data accumulate, allowing for up-to-date analysis of growing data with only a small accuracy cost relative to the static setting. Our work gives the first private algorithms for answering adaptive queries in the dynamic setting.

## 1.1 Our results

Here we outline our two sets of results for adaptive analysis of dynamically growing databases. Throughout the paper, we refer to the setting in which a database of $n$ elements from a universe of size $N$ is fixed for the life of the analysis as the *static setting*, and we refer to the setting in which a database is accumulating new data entries while the analysis is ongoing as the *dynamic setting*. We use the standard definition of differential privacy, presented formally along with other notation in the preliminaries.

**Adaptive linear queries for growing databases.** Our first result is a novel modification of the private multiplicative weights (PMW) algorithm [HR10], a broadly useful algorithm for privately answering an adaptive stream of linear queries. The static PMW algorithm works by maintaining a public histogram that reflects the current estimate of the database given all previously answered queries. It categorizes incoming queries as either easy or hard, updating the histogram and suffering significant privacy loss only for the hard queries. The number of hard queries is bounded using a potential argument, where the potential is defined as the relative entropy between the true database and the public histogram. This quantity is initially bounded, decreases by a substantial amount after every hard query, and never increases.

The main challenge in adapting PMW to the dynamic setting is that new data increase the number of opportunities for privacy loss, harming the privacy-accuracy tradeoff. If we run static PMW on a growing database, the previous potential argument fails because the relative entropy between the database and the public histogram can increase as new data arrive. In the worst case, PMW can learn the true database with high accuracy (using many hard queries), and then adversarial data growth will change the composition of the database dramatically, essentially requiring the maximum possible number of additional hard queries to retain the same accuracy.

We modify PMW so that when new data arrive, the algorithm adds a uniform distribution to the public histogram and re-normalizes. This leads to no additional privacy loss and requires no assumptions on the actual distribution of the new data. This technique defends against adversarial data growth that could dramatically increase the relative entropy between the public histogram and the true database incorporating the new data, allowing us to maintain the accuracy guarantee of the static setting through unbounded data growth. Specifically, static PMW works on a fixed database of size $n$ and answers $k$ linear queries. In comparison, our modification for growing databases (PMWG) works on a database of starting size $n$ and at each time step when the database is size $t \geq n$ answers up to $\kappa \cdot \exp(\sqrt{t/n})$ queries.

**Theorem 1** (Informal version of Theorem 5). PMWG is $\epsilon$-differentially private and for any stream with up to $\kappa \cdot \exp(\sqrt{t/n})$ queries at each time $t \geq n$ incurs additive error at most $\alpha = O((\frac{\log N \log \kappa}{\epsilon n})^{1/3})$ for all queries with high probability.

This error bound is tight with respect to static PMW, which incurs additive error $O((\frac{\log N \log k}{\epsilon n})^{1/3})$ for only $k$ total queries. This is somewhat surprising, given that the dynamic setting is strictly harder than the static setting. Even on just the first time step when $t = n$, PMWG must answer $\kappa$ queries on a database of size $n$, and it achieves the same error guarantee on those queries as static PMW. Static PMW terminates at this point, while PMWG will answer another $\kappa \cdot \exp(\sqrt{(n+1)/n})$ queries at the next time step and will continue answering queries as the database grows.

In the process of proving Theorem 1, we develop extensions of several static differentially private algorithms to the dynamic setting, which may be of independent interest for future work on the

design of differentially private algorithms for growing databases. These algorithms are presented in Appendix C.

**General transformations of static algorithms into algorithms for growing databases.** Our second set of results consists of two methods, BBSCHEDULER and BBIMPROVER, for generically transforming a black box algorithm that is private and accurate in the static setting into an algorithm that is private and accurate in the dynamic setting. BBSCHEDULER reruns the black box algorithm every time the database increases in size (starting from $n$) by a small multiplicative factor, and it provides privacy and accuracy guarantees that are independent of the total number of queries and the current database size (Theorem 27). BBSCHEDULER instantiates each successive run of the black box algorithm with an exponentially shrinking privacy parameter to achieve any desired total privacy loss. The privacy parameter's decay is tied to database growth so that the two scale together, yielding a time-independent accuracy guarantee. We instantiate this scheduler using the SMALLDB algorithm for answering linear queries as a black box (Corollary 10).

Our second transformation, BBIMPROVER, runs the black box every time a new entry is added to the database. As with BBSCHEDULER, the privacy parameter decreases for successive calls to the black box, but in this case this shrinking eventually dominates the database growth to yield accuracy guarantees that improve as more data accumulate. This algorithm is well-suited for problems where data points are sampled from a distribution, where one would expect the accuracy guarantees of static analysis to improve with the size of the sample. We apply this scheduler to private empirical risk minimization (ERM) algorithms to output classifiers with generalization error that improves as the training database grows (Table 3).

The following informal theorem statement summarizes our results for BBSCHEDULER (Theorem 27) and BBIMPROVER (Theorem 29). Taken together, these results show that almost any private and accurate algorithm can be rerun at appropriate points of data growth with minimal loss of accuracy, even when data growth is unbounded.

**Theorem 2** (Informal). Let $\mathcal{M}$ be an $\epsilon$-differentially private algorithm that for some constant $p$ incurs additive error $\alpha = \tilde{O}\left(\left(\frac{1}{\epsilon n}\right)^p\right)$ for all queries with high probability. Then,

1. BBSCHEDULER running $\mathcal{M}$ is $\epsilon$-differentially private and incurs additive error $\alpha = \tilde{O}\left(\left(\frac{1}{\epsilon n}\right)^{p/(2p+1)}\right)$ for all queries with high probability.

2. BBIMPROVER running $\mathcal{M}$ is $(\epsilon, \delta)$-differentially private and incurs additive error $\alpha_t = \tilde{O}\left(\left(\frac{\sqrt{\log(1/\delta)}}{\epsilon\sqrt{t}}\right)^p\right)$ for all queries at time $t$ for all $t \geq n$ with high probability.

## 1.2 Related Work

Differential privacy for growing databases has been studied for a limited class of problems. We summarize the relationship between our work and the most relevant previous work in Table 1. Both [DNPR10] and [CSS11] adapted the notion of differential privacy to streaming environments in a setting where each entry in the database is a single bit, and bits arrive one per unit time. [DNPR10] and [CSS11] design differentially private algorithms for an analyst to maintain an approximately accurate count of the number 1-bits seen thus far in the stream. This technique was later extended by [ST13] to maintain private sums of real vectors arriving online in a stream. We note that both of these settings correspond to only a single query repeatedly asked on a dynamic database, precluding meaningful adaptive analysis. In contrast, we consider the much richer class of *linear queries*, including $2^{|\mathcal{X}|}$ counting queries, allowing for adaptive analysis of a dynamically growing database.

Our setting also resembles the online learning setting, but differs in that we are interested in per-round accuracy bounds, rather than regret bounds. We discuss this connection in more detail in Appendix A, along with background on private adaptive analysis of a static databases.

## 2 Preliminaries

All algorithms in this paper take as inputs a database over some fixed data universe $\mathcal{X}$ of finite size $N$. Our algorithms and analyses represent a finite database $D \in \mathcal{X}^n$ equivalently as a fractional

Table 1: Asymptotic accuracy guarantees for answering adaptive linear queries

| | Work | Database | Queries | Accuracy |
|---|---|---|---|---|
| Previous work | SmallDB [BLR08] | static | linear queries, non-adaptive | fixed |
| | PMW [HR10] | static | linear queries, adaptive | fixed |
| | Counting bits [DNPR10, CSS11] | dynamic | one fixed query, non-adaptive | improving as database grows |
| Our work | PMWG | dynamic | linear queries, adaptive | fixed |
| | BBScheduler | dynamic | any queries, adaptive | fixed |
| | BBImprover | dynamic | any queries, adaptive | improving as database grows |

histogram $x \in \Delta(\mathcal{X}) \subseteq \mathbb{R}^N$, where $x^i$ is the fraction of the database of type $i \in [N]$. When we say a database $x \in \Delta(\mathcal{X})$ has size $n$, this means that for each $i \in [N]$ there exists some $n_i \in \mathbb{N}$ such that $x^i = n_i/n$.

If an algorithm operates over a single fixed database, we refer to this as the static setting. In the dynamic setting, algorithms operate over a *stream of databases*, defined as a sequence of databases $X = \{x_t\}_{t \geq n}$ starting with a database $x_n$ of size $n$ at time $t = n$ and increasing by one data entry per time step so that $t$ always denotes both a time and the size of the database at that time. Our dynamic algorithms also take a parameter $n$, which denotes the starting size of the database.

We consider algorithms that answer real-valued queries $f : \mathbb{R}^N \to \mathbb{R}$ with particular focus on *linear queries*. A linear query assigns a weight to each entry depending on its type and averages these weights over the database. We can interpret a linear query as a vector $f \in [0,1]^N$ and write the answer to the query on database $x \in \Delta(\mathcal{X})$ as $\langle f, x \rangle$, $f(x)$, or $x(f)$, depending on context. For $f$ viewed as a vector, $f^i$ denotes the $i$th entry. We note that an important special case of linear queries are counting queries, which calculate the proportion of entries in a database satisfying some boolean predicate over $\mathcal{X}$.

Many of the algorithms we study allow queries to be chosen *adaptively*, i.e., the algorithm accepts a stream of queries $F = \{f_j\}_{j=1}^k$ where the choice of $f_{j+1}$ can depend on the previous $j-1$ queries and answers. For the dynamic setting, we doubly index a stream of queries as $F = \{f_{t,:}\}_{t \geq n} = \{\{f_{t,j}\}_{j=1}^{\ell_t}\}_{t \geq n}$ so that $t$ denotes the size of the database at the time $f_{t,j}$ is received and $j = 1, \ldots, \ell_t$ indexes the queries received when the database is size $t$.

The algorithms studied produce outputs of various forms. To evaluate accuracy, we assume that an output $y$ of an algorithm for query class $\mathcal{F}$ (possibly specified by an adaptively chosen query stream) can be interpreted as a function over $\mathcal{F}$, i.e., we write $y(f)$ to denote the answer to $f \in \mathcal{F}$ based on the mechanism's output. We seek to develop mechanisms that are accurate in the following sense.

**Definition 1** (Accuracy in the static setting). For $\alpha, \beta > 0$, an algorithm $\mathcal{M}$ is $(\alpha, \beta)$-*accurate* for real query class $\mathcal{F}$ if for any input database $x \in \Delta(\mathcal{X})$, the algorithm outputs $y$ such that $|f(x) - y(f)| \leq \alpha$ for all $f \in \mathcal{F}$ with probability at least $1 - \beta$.

In the dynamic setting, accuracy must be with respect to the current database, and the bounds may be parametrized by time.

**Definition 2** (Accuracy in the dynamic setting). For $\alpha_n, \alpha_{n+1}, \ldots > 0$ and $\beta > 0$, an algorithm $\mathcal{M}$ is $(\{\alpha_t\}_{t \geq n}, \beta)$-accurate for query stream $F = \{f_{t,:}\}_{t \geq n}$ if for any input data stream $X = \{x_t\}_{t \geq n}$, the algorithm outputs $y$ such that $|f_{t,j}(x_t) - y(f_{t,j})| \leq \alpha_t$ for all $f_{t,j} \in F$ with probability at least $1 - \beta$.

## 2.1 Differential privacy and composition lemmas

Differential privacy in the static setting requires that an algorithm produce similar outputs on *neighboring databases* $x \sim x'$, which differ by a single entry. In the dynamic setting, differential privacy requires similar outputs on *neighboring database streams* $X, X'$ that satisfy that for some $t \geq n$,

$x_\tau = x'_\tau$ for $\tau = n, \ldots, t-1$ and $x_\tau \sim x'_\tau$ for $\tau = t, t+1, \ldots$. In the definition below, a pair of *neighboring inputs* refers to a pair of neighboring databases in the static setting or a pair of neighboring database streams in the dynamic setting. We note that in the dynamic setting, an element in $\mathrm{Range}(\mathcal{M})$ is an entire (potentially infinite) transcript of outputs that may be produced by $\mathcal{M}$.

**Definition 3** (Differential privacy [DMNS06])**.** For $\epsilon, \delta > 0$, an algorithm $\mathcal{M}$ is $(\epsilon, \delta)$-*differentially private* if for any pair of neighboring inputs $x, x'$ and any subset $S \subseteq \mathrm{Range}(\mathcal{M})$,

$$\Pr[\mathcal{M}(x) \in S] \leq e^\epsilon \cdot \Pr[\mathcal{M}(x') \in S] + \delta.$$

When $\delta = 0$, we will say that $\mathcal{M}$ is $\epsilon$-differentially private.

Differential privacy is typically achieved by adding random noise that scales with the *sensitivity* of the computation being performed. The sensitivity of any real-valued query $f : \Delta(\mathcal{X}) \to \mathbb{R}$ is the maximum change in the query's answer due to the change of a single entry in the database, denoted $\Delta_f = \max_{x \sim x'}|f(x) - f(x')|$. Note that a linear query on a database of size $n$ has sensitivity $1/n$.

The following composition theorems quantify how the privacy guarantee degrades as additional computations are performed on a database.

**Theorem 3** (Basic composition, [DMNS06])**.** Let $\mathcal{M}_i$ be an $\epsilon_i$-differentially private algorithm for all $i \in [k]$. Then the composition $\mathcal{M}$ defined as $\mathcal{M}(x) = (\mathcal{M}_i(x))_{i=1}^k$ is $\epsilon$-differentially private for $\epsilon = \sum_{i=1}^k \epsilon_i$.

**Theorem 4** (CDP composition, Corollary of [BS16])**.** Let $\mathcal{M}_i$ be a $\epsilon_i$-differentially private algorithm for all $i \in [k]$. Then the composition of $\mathcal{M}$ defined as $\mathcal{M}(x) = (\mathcal{M}_i(x))_{i=1}^k$ is $(\epsilon, \delta)$-differentially private for $\epsilon = \frac{1}{2}(\sum_{i=1}^k \epsilon_i^2) + \sqrt{2(\sum_{i=1}^k \epsilon_i^2)\log(1/\delta)}$. In particular, for $\delta \leq e^{-1}$ and $\sum_{i=1}^T \epsilon_i^2 \leq 1$, we have $\epsilon \leq 2\sqrt{(\sum_{i=1}^k \epsilon_i^2)\log(1/\delta)}$.

## 3 Adaptive linear queries for growing databases

In this section we show how to modify the static private multiplicative weights (PMW) algorithm [HR10] for the dynamic setting to allow for private and accurate adaptive analysis of a growing database. Static PMW answers an adaptive stream of linear queries while maintaining a public histogram $y$ reflecting the current estimate of the static database $x$ given all previously answered queries. Critical to the performance of the algorithm is that it uses the public histogram to categorize incoming queries as either easy or hard, and it updates the histogram after hard queries in a way that moves it closer to a correct answer on that query. The number of hard queries is bounded using a potential argument, where potential is defined as the relative entropy between the database and the public histogram, i.e., $\mathrm{RE}\,(x||y) = \sum_{i \in [N]} x^i \log(x^i/y^i)$. This quantity is initially bounded, decreases by a substantial amount after every hard query, and never increases. However, this argument does not extend to the dynamic setting because the potential can increase with the arrival of new data. We instead modify the algorithm so the public histogram updates in response to new data arrivals as well as hard queries. This modification allows us to suffer only constant loss in accuracy per query relative to the static setting, while maintaining this accuracy through unbounded data growth and a growing query budget at each stage of growth. Table 2 compares our results to the static setting.

We remark that PMW runs in time linear in the data universe size $N$. If the incoming data entries are drawn from a distribution that satisfies a mild smoothness condition, a compact representation of the data universe can significantly reduce the runtime [HR10]. The same idea applies to our modification of PMW for the dynamic setting without requiring new technical tools.

### 3.1 Private multiplicative weights for growing databases (PMWG)

Our algorithm for PMW for growing databases (PMWG) is given as Algorithm 1 in Appendix B. We give an overview here to motivate our main results. The algorithm takes as inputs a data stream $X = \{x_t\}_{t \geq n}$ and an adaptively chosen query stream $F = \{\{f_{t,j}\}_{j=1}^{\ell_t}\}_{t \geq n}$. It also accepts privacy and accuracy parameters $\epsilon, \delta, \alpha > 0$, although in this section we consider the case that $\delta = 0$.

The algorithm maintains a fractional histogram $y$ over $\mathcal{X}$, where $y_{t,j}$ denotes the histogram after the $j$th query at time $t$ has been processed. This histogram is initialized to uniform, i.e., $y_{n,0}^i = 1/N$

for all $i \in [N]$. As with static PMW, when a query is deemed hard, our algorithm performs a multiplicative weights update of $y$ with learning rate $\alpha/6$. As an extension of the static case, we also update the weights of $y$ when a new data entry arrives to reflect a data-independent prior belief that data arrive from a uniform distribution, i.e., for all $t > n, i \in [N]$, $y_{t,0}^i = \frac{t-1}{t} y_{t-1,\ell_{t-1}}^i + \frac{1}{t}\frac{1}{N}$. It is important to note that a multiplicative weights update depends only on the noisy answer to a hard query as in the static case, and the uniform update only depends on the knowledge that a new entry arrived, so this histogram can be thought of as public.

As in static PMW, we determine hardness using a numeric sparse subroutine. As part of our proof, we adapt the Numeric Sparse and the underlying Above Threshold algorithms of [DNR$^+$09] to the dynamic setting. The proofs for our dynamic versions of these algorithms are in Appendix C and may be of independent interest for future work in the design of private algorithms for growing databases.

We now present our main result for PMWG, Theorem 5. We sketch its proof here and give the full proof in Appendix B.1. Whereas the accuracy results for static PMW are parametrized by the total allowed queries $k$, our noise scaling means our algorithm can accommodate more and more queries as new data continue to arrive. Our accuracy result is with respect to a query stream respecting a query budget. This budget increases at each time $t$ by a quantity increasing exponentially with $\sqrt{t}$, and it is parametrized by some time-independent $\kappa \geq 1$, which is somewhat analogous to the total query budget $k$ in static PMW. This theorem tells us that PMWG can accommodate $\mathrm{poly}(\kappa)$ queries on the original database. Since $\kappa$ degrades accuracy logarithmically, this means we can accurately answer exponentially many queries before any new data arrive. In particular, our accuracy bounds are tight with respect to the static setting[2], and we maintain this accuracy through unbounded data growth, subject to a generous query budget specified by the theorem's bound on $\sum_{\tau=n}^{t} \ell_\tau$.

**Theorem 5.** The algorithm $\mathrm{PMWG}(X, F, \epsilon, 0, \alpha, n)$ is $(\epsilon, 0)$-differentially private, and for any time-independent $\kappa \geq 1$ and $\beta > 0$ it is $(\alpha, \beta)$-accurate for any query stream $F$ such that $\sum_{\tau=n}^{t} \ell_\tau \leq \kappa \sum_{\tau=n}^{t} \exp(\frac{\alpha^3 \epsilon \sqrt{n\tau}}{C \log(Nn)})$ for all $t \geq n$ and sufficiently large constant $C$ as long as $N \geq 3, n \geq 21$ and

$$\alpha \geq C \left( \frac{\log(Nn) \log(\kappa n/\beta)}{n\epsilon} \right)^{1/3}.$$

*Proof sketch.* The proof hinges on showing that we do not have to answer too many hard queries, even as the composition of the database changes with new data, which can increase the relative entropy between the database and the public histogram. We first show that our new public histogram update rule bounds this relative entropy increase (Lemma 6), and then our bound on the number of hard queries suffers accordingly relative to static PMW (Corollary 7).

**Lemma 6.** Let $x, y, \bar{x}, \bar{y} \in \Delta(\mathcal{X})$ be databases of size $t, t, t+1, t+1$, respectively, where $\bar{x}$ is obtained by adding one entry to $x$ and $\bar{y}^i = \frac{t}{t+1} y^i + \frac{1}{(t+1)N}$ for $i \in [N]$. Then,

$$\mathrm{RE}\left(\bar{x}||\bar{y}\right) - \mathrm{RE}\left(x||y\right) \leq \frac{\log N}{t+1} + \frac{\log t}{t+1} + \log(\frac{t+1}{t}).$$

The corollary below comes from a straightforward modification of the proof on the bound on hard queries in static PMW using the result above.

**Corollary 7.** If the numeric sparse subroutine returns $\alpha/3$-accurate answers for each query for a particular run of PMWG, then the total number of hard queries answered by any time $t \geq n$ is

$$\sum_{\tau=n}^{t} h_\tau \leq \frac{36}{\alpha^2}(\log N + \sum_{\tau=n+1}^{t} \frac{\log(N)}{\tau} + \frac{\log(\tau-1)}{\tau} + \log(\frac{\tau}{\tau-1})).$$

With this corollary, we separately prove privacy and accuracy (Theorems 11 and 12) in terms of the noise function $\xi$, which yield our desired result when instantiated with the $\xi$ specified by Algorithm 1. As with static PMW, the only privacy is leaked by the numeric sparse subroutine. Privacy loss depends in the usual ways on the noise parameter, query sensitivity, and number of hard queries, although in our setting both the noise parameter and query sensitivity change over time. $\square$

Table 2: Asymptotic accuracy guarantees for answering adaptive linear queries

| Work | Setting | Accuracy for $(\epsilon, 0)$-DP | Accuracy for $(\epsilon, \delta)$-DP |
|------|---------|---------------------------------|--------------------------------------|
| [HR10] | Static | $\left(\frac{\log N \log(k/\beta)}{\epsilon n}\right)^{1/3}$ | $\left(\frac{\log^{1/2} N \log(k/\beta) \log(1/\delta)}{\epsilon n}\right)^{1/2}$ |
| This work | Dynamic | $\left(\frac{\log(Nn) \log(\kappa n/\beta)}{\epsilon n}\right)^{1/3}$ | $\left(\frac{\log^{1/2}(Nn) \log(\kappa n/\beta) \log^{1/2}(1/\delta)}{\epsilon n}\right)^{1/2}$ |

After the proof of the above theorem in Appendix B.1, Theorem 16 generalizes PMWG as specified by Equation (B.5). This generalization leaves a free parameter in the noise function $\xi$ used by the subroutine, allowing one to trade off between accuracy and a query budget that increases more with time. See Observation 17.

We remark that we can tighten our accuracy bounds if we allow $(\epsilon, \delta)$-differential privacy and use CDP composition [BS16]. These results are proven in Appendix B.2 and included informally in Table 2.

**Theorem 8.** The algorithm $\text{PMWG}(X, F, \epsilon, \delta, \alpha, n)$ is $(\epsilon, \delta)$-differentially private for any $\epsilon \in (0, 1], \delta \in (0, e^{-1})$, and for any time-independent $\kappa \geq 1$ and $\beta \in (0, 2^{-15/2})$ it is $(\alpha, \beta)$-accurate for any query stream $F$ such that $\sum_{\tau=n}^{t} \ell_\tau \leq \kappa \sum_{\tau=n}^{t} \exp(\frac{\alpha^2 \epsilon \sqrt{n\tau}}{C \log^{1/2}(Nn) \log^{1/2}(1/\delta)})$ for all $t \geq n$ and sufficiently large constant $C$ as long as $N \geq 3, n \geq 17$ and

$$\alpha \geq C \left(\frac{\log^{1/2}(Nn) \log^{1/2}(1/\delta) \log(\kappa n/\beta)}{n\epsilon}\right)^{1/2} .$$

## 4 General transformations from static to dynamic settings

In this section, we give two schemes for answering a stream of queries on a growing database, given black box access to a differentially private algorithm for the static setting.[3] In Section 4.1, we describe an algorithm BBSCHEDULER for scheduling repeated runs of a static algorithm. BBSCHEDULER runs an underlying offline mechanism with exponentially decreasing frequency and offers the same accuracy guarantee at every point in data growth. We instantiate BBSCHEDULER with the SmallDB algorithm as an illustrative example. In Section 4.2, we describe a second algorithm BBIMPROVER, which runs an underlying mechanism at every time step. Its results are initially inferior but improve over BBSCHEDULER with sufficient data growth. This result is well-suited for problems where data points are sampled from a distribution, where one would expect the accuracy guarantees of static analysis to improve with the size of the sample. We showcase our result by applying it to solve private empirical risk minimization on a growing database. We formalize these algorithms and give privacy and accuracy guarantees in full generality in Appendix D.

### 4.1 Fixed accuracy as data accumulate

In this section, we give results for using a private and accurate algorithm for the static setting as a black box to solve the analogous problem in the dynamic setting. Our general purpose algorithm BBSCHEDULER treats a static algorithm as a black box endowed with privacy and accuracy guarantees, and it reruns the black box whenever the database grows by a small multiplicative factor. This schedule can be applied to any algorithm that satisfies $\epsilon$-differential privacy and $(\alpha, \beta)$-accuracy for $\alpha$ of a certain form as specified in Definition 4 below.

**Definition 4** ($(p, g)$-black box)**.** An algorithm $\mathcal{M}(x_n, \epsilon, \alpha, \beta, n)$ is a $(p, g)$-black box for a class of linear queries $\mathcal{F}$ if it is $(\epsilon, 0)$-differentially private and with probability $1 - \beta$ it outputs $y : \mathcal{F} \to \mathbb{R}$ such that $|y(f) - x_n(f)| \leq \alpha$ for every $f \in \mathcal{F}$ when $\alpha \geq g \cdot (\frac{\log(1/\beta)}{\epsilon n})^p$ for some $g$ that is independent of $\epsilon, n, \beta$.

The parameter $g$ captures dependence on domain-specific parameters that affect accuracy of the black box algorithm, such as the dependence on $\log N$ for SMALLDB. If these other parameters

are constant, then $\alpha = \Theta((\frac{\log(1/\beta)}{\epsilon n})^p)$. As a concrete example, see Corollary 10 and surrounding discussion as an instantiation of BBSCHEDULER with the SMALLDB algorithm as a black box.

Our generic algorithm BBSCHEDULER runs the black box $\mathcal{M}(x_{t_i}, \epsilon_i, \alpha_i, \beta_i, t_i)$ at times $\{t_i\}_{i=0}^{\infty}$ for $t_i = (1+\gamma)^i n$ with parameters as listed below and receives output $y_i$. Upon receipt of query $f_{t,j}$ for $t \in [t_i, \ldots, t_{i+1}]$, we output $y_i(f_{t,j})$. We give the $\delta = 0$ case below; the full algorithm including parameter settings for the $\delta > 0$ case is presented in Appendix D.

$$\gamma = g^{\frac{1}{2p+1}} \left(\frac{\log(1/\beta)}{\epsilon n}\right)^{\frac{p}{2p+1}}, \quad \epsilon_i = \frac{\gamma^2(i+1)}{(1+\gamma)^{i+2}}\epsilon, \quad \alpha_i = g\left(\frac{\log(1/\beta)}{\epsilon_i(1+\gamma)^i n}\right)^p, \quad \beta_i = \left(\frac{\beta}{1+\beta}\right)^{i+1}$$

There are two key technical properties that allow this result to hold. First, since the epochs are exponentially far apart, the total privacy loss from multiple calls to $\mathcal{M}$ is not too large. Second, each data point added to a database of size $t$ can only change a linear query by roughly $\frac{1}{t}$, so since a database grows by $\gamma t_i$ in epoch $i$, an answer to a query at the end of epoch $i$ using $y_i$ incurs at most $\gamma$ extra additive error relative to a query issued at time $t_i$. We now state our main result for BBSCHEDULER, including the result for $\delta > 0$:

**Theorem 9.** Let $\mathcal{M}$ be a $(p, g)$-black box for query class $\mathcal{F}$. Then for any database stream $X$ and stream of linear queries $F$ over $\mathcal{F}$, BBSCHEDULER$(X, F, \mathcal{M}, \epsilon, \delta, \beta, n, p, g)$ is $(\epsilon, \delta)$-differentially private for $\epsilon < 1$ and $(\alpha, \beta)$-accurate for sufficiently large constant $C$ and

$$\alpha \geq \begin{cases} Cg^{\frac{1}{2p+1}} \cdot \left(\frac{\log(1/\beta)}{\epsilon n}\right)^{\frac{p}{2p+1}} & \text{if } \delta = 0 \\ Cg^{\frac{1}{1.5p+1}} \cdot \left(\frac{\sqrt{\log(1/\delta)}\log(1/\beta)}{\epsilon n}\right)^{\frac{p}{1.5p+1}} & \text{if } \delta > 0 \end{cases}.$$

For concreteness, we instantiate this general result with SMALLDB [BLR08], a differentially private algorithm for generating a synthetic database $y$ that closely approximates a true database $x$ on a every query from some fixed set $\mathcal{F}$ of $k$ linear queries. Specifically, SMALLDB outputs some $y : \mathcal{F} \to \mathbb{R}$ such that $|y(f) - x(f)| \leq \alpha$ for every $f \in \mathcal{F}$ when $\alpha \geq C \left(\frac{\log N \log k + \log(1/\beta)}{\epsilon n}\right)^{1/3}$. SMALLDB is thus a $(1/3, C(\log N \log k)^{1/3})$-black box for an arbitrary set of $k$ linear queries over a data universe of size $N$, and so we have the following corollary of Theorem 27.

**Corollary 10.** BBSCHEDULER instantiated with SMALLDB is $\epsilon$-differentially private and can answer all queries in $F$ with $(\alpha, \beta)$-accuracy for sufficiently large constant $C$ and

$$\alpha \geq C \left(\frac{\log N \log |\mathcal{F}| \log(1/\beta)}{\epsilon n}\right)^{1/5}.$$

## 4.2 Improving accuracy as data accumulate

In some applications it is more natural for accuracy bounds to improve as the database grows. For instance, in empirical risk minimization (ERM), we expect to be able to find classifiers with diminishing empirical risk, which implies diminishing generalization error.

We can extend our black box scheduler framework to allow for accuracy guarantees that improve as data accumulate. Like our first scheduler, our new algorithm BBIMPROVER takes in a private and accurate static black box $\mathcal{M}$. Unlike the first scheduler, it reruns $\mathcal{M}$ on the current database at every time step. The algorithm no longer incurs accuracy loss from ignoring new data points mid-epoch because it runs $\mathcal{M}$ at every time step. However, this also means that privacy loss will accumulate much faster because more computations are being composed. To combat this and achieve overall privacy loss $\epsilon$, each run of $\mathcal{M}$ will have increasingly strict (i.e., smaller) privacy parameter $\epsilon_t$. The additional noise needed to preserve privacy will overpower the improvements in accuracy until the database grows sufficiently large ($t \gg n^2$), when the accuracy of BBIMPROVER will surpass the comparable fixed accuracy guarantee of BBSCHEDULER. Our BBIMPROVER algorithm and general results (Theorem 29) are presented in Appendix D. We also instantiate BBIMPROVER with various private ERM algorithms in Theorem 31 in Appendix E.

### Acknowledgements

R.C. and S.K. supported in part by a Mozilla Research Grant. K.L. supported in part by NSF grant IIS-1453304. U.T. supported in part by NSF grants CCF-24067E5 and CCF-1740776, and by a Georgia Institute of Technology ARC fellowship.

## Footnotes

[2]This tightness claim assumes $n = O(\mathrm{poly}(N))$. We think of PMW as being useful in this setting when the data universe is large relative to the size of the database, otherwise an analyst could learned the dataset more accurately with $N \ll n$ counting queries using output perturbation.

[3] For ease of presentation, we restrict our results to accuracy of real-valued queries, but the algorithms we propose could be applied to settings with more general notions of accuracy or to settings where the black box algorithm itself can change across time steps, adding to the adaptivity of this scheme.

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
