[Supplementary Material]

# A  Other Related Work

In this section, we describe related work in other settings that are similar to the dynamic setting.

**Online Learning.** Our setting of dynamically growing databases is also closely related to online learning, where a learner plays a game with an adversary over many rounds. In each round $t$, the adversary first gives the learner some input, then the learner chooses an action $a_t$ and receives loss function $\mathcal{L}_t$ chosen by the adversary, and experiences loss $\mathcal{L}_t(a_t)$. There is a vast literature on online learning, including several works on differentially private online learning [JKT12, ST13, AS17]. In those settings, a database is a sequence of loss functions, and neighboring databases differ on a single loss function. While online learning resembles the dynamic database setting, there are several key differences. Performance bounds in the online setting are in terms of regret, which is a cumulative error term. On the other hand, we seek additive error bounds that hold for all of our answers. Such bounds are not possible in general for online learning, since the inputs are adversarial and the true answer is not known. In our case, we can achieve such bounds because even though queries are presented adversarially, we have access to the query's true answer. Instead of a cumulative error bound, we manage a cumulative privacy budget.

**Private Adaptive Analysis of a Static Database.** If we wish to answer multiple queries on the same database by independently perturbing each answer, then the noise added to each answer must scale linearly with the number of queries to maintain privacy, meaning only $O(n)$ queries can be answered with meaningful privacy and accuracy. If the queries are known in advance, however, [BLR08] showed how to answer exponentially many queries relative to the database size for fixed $\epsilon$ and $\alpha$. Later, Private Multiplicative Weights (PMW) [HR10] achieved a similar result in the interactive setting, where the analyst can adaptively decide which queries to ask based on previous outputs. Their accuracy guarantee is close to the sample error of $\sqrt{\log k/n}$. A recent line of work [DFH+15, CLN+16, BNS+16] showed deep connections between differential privacy and adaptive data analysis of a static database. Our results would allow analysts to apply these tools on dynamically growing databases.

# B Analysis of PMW for Growing Databases

We present PMWG, the modification of PMW for growing databases, described in Section 3 formally in Algorithm 1. We separately prove privacy and accuracy in terms of the internal noise function $\xi$, which depends on the parameters of the algorithm. We then instantiate these theorems with our particular choice of $\xi$ to prove the our accuracy bound for the $(\epsilon, 0)$-private version of the algorithm (Theorem 5 in the body). Later, we describe how to generalize our algorithm by adding a parameter for the noise function (Equation (B.5)), which allows us to trade accuracy for a larger query budget. Finally, we use CDP to give our $(\epsilon, \delta)$-results for PMWG (Theorem 8 in the body).

---

**Algorithm 1** PMWG$(X, F, \epsilon, \delta, \alpha, n)$

---

**if** $\delta = 0$ **then**
    Let $\xi_t \leftarrow \frac{\alpha^2 n^{1/2}}{162 \log(Nn)} \epsilon t^{1/2}$ for $t \geq n$
**else**
    Let $\xi_t \leftarrow \frac{\alpha n^{1/2}}{48 \log^{1/2}(Nn) \log^{1/2}(1/\delta)} \epsilon t^{1/2}$ for $t \geq n$
**end if**
Start NSG$(X, \cdot, 2\alpha/3, \{\xi_t\}_{t \geq n})$            # Initialize subroutine
Let $y_{n,0}^i \leftarrow 1/N$ for $i \in [N]$               # Public histogram
Let $h_t \leftarrow 0$ for $t \geq n$                    # Hard query counters
Let $b_t \leftarrow \frac{\log N}{t} + \frac{\log(t-1)}{t} + \log \frac{t}{t-1}$ for $t \geq n + 1$    # Hard query bounds
**for** each incoming query $f_{t,j}$ **do**
    **if** last query was at time $t' < t$ **then**
        Set $y_{t,0}^i = \frac{t'}{t} y_{t',\ell_{t'}}^i + \frac{t-t'}{t} \frac{1}{N}$              # Uniform update
    **end if**
    Set $f'_{t,2j-1} = f_{t,j} - f_{t,j}(y_{t,j-1}), f'_{t,2j} = f_{t,j}(y_{t,j-1}) - f_{t,j}$
    Receive $a'_{t,2j-i}, a'_{t,2j}$ from NSG on $f'_{t,2j-1}, f'_{t,2j}$     # Check hardness
    **if** $a'_{t,2j-1} = \perp$ and $a'_{t,2j} = \perp$ **then**
        Set $y_{t,j} = y_{t,j-1}$
        Set $a_{t,j} = f_j(y_{t,j})$                # Compute easy query answer
    **else**
        Set $h_t = h_t + 1$
        **if** $\sum_{\tau=n}^t h_\tau > \frac{36}{\alpha^2} \left( \log N + \sum_{\tau=n+1}^t b_\tau \right)$ **then**
            **return** $\perp$                # Hard query budget exceeded
        **end if**
        **if** $a'_{t,2j-1} \in \mathbb{R}$ **then**
            Set $a_{t,j} = f_{t,j}(y_{t,j-1}) + a'_{t,2j-1}$
        **else**
            Set $a_{t,j} = f_{t,j}(y_{t,j-1}) - a'_{t,2j}$
        **end if**                    # Compute hard query answer
        **if** $a_{t,j} < f_{t,j}(y_{t,j-1})$ **then**
            Set $r_{t,j} = f_{t,j}$
        **else**
            Set $r_{t,j} = 1 - f_{t,j}$
        **end if**
        Set $\hat{y}_{t,j}^i = \exp\left(-\frac{\alpha}{6} r_{t,j}^i\right) y_{t,j-1}^i$ for $i \in [N]$
        Set $y_{t,j}^i = \frac{\hat{y}_{t,j}^i}{\sum_{i' \in [N]} \hat{y}_{t,j}^{i'}}$ for $i \in [N]$       # MW update
    **end if**
**end for**

---

## B.1 $(\epsilon, 0)$-DP for PMWG

The $(\epsilon, 0)$-privacy guarantee of PMWG follows from the NSG analysis in Theorem 24 as well as Lemma 6 bounding the entropy increase due to new data and Corollary 7 bounding the number of hard queries received at any given time. We first prove Lemma 6 and then state and prove privacy.

**Lemma 6.** Let $x, y, \bar{x}, \bar{y} \in \Delta(\mathcal{X})$ be databases of size $t, t, t+1, t+1$, respectively, where $\bar{x}$ is obtained by adding one entry to $x$ and $\bar{y}^i = \frac{t}{t+1}y^i + \frac{1}{(t+1)N}$ for $i \in [N]$. Then,

$$\mathrm{RE}\,(\bar{x}||\bar{y}) - \mathrm{RE}\,(x||y) \leq \frac{\log N}{t+1} + \frac{\log t}{t+1} + \log(\tfrac{t+1}{t}).$$

*Proof.* We partition indices $i \in [N]$ into two sets $L, H$, where $i \in L$ whenever $y^i \leq \frac{1}{tN}$. For each $S \subseteq [N]$, use $x_S$ to denote $\sum_{i \in S} x^i$. Then, by $\bar{y}^i \geq \frac{1}{(t+1)N}$ for all $i$,

$$
\begin{aligned}
\sum_{i \in L} \left(\bar{x}^i \log(1/\bar{y}^i) - x^i \log(1/y^i)\right) &\leq \sum_{i \in L} \left(\bar{x}^i \log(N(t+1)) - x^i \log(1/y^i)\right) \\
&\leq \sum_{i \in L} \left(\bar{x}^i \log(N(t+1)) - x^i \log(tN)\right) \\
&= \sum_{i \in L} \left(\bar{x}^i \log(Nt) + \bar{x}^i \log(\frac{t+1}{t}) - x^i \log(tN)\right) \\
&= \sum_{i \in L} (\bar{x}^i - x^i) \log(Nt) + \bar{x}_L \log(\frac{t+1}{t}) \\
&\leq \sum_{i \in L} \max\{(\bar{x}^i - x^i), 0\} \log(Nt) + \bar{x}_L \log(\frac{t+1}{t})
\end{aligned}
$$

The last inequality is by ignoring the term $i \in L$ with $\bar{x}^i < x^i$. Next, we use $\bar{y}^i \geq \frac{t}{t+1}y^i$ to get

$$
\begin{aligned}
\sum_{i \in H} \left(\bar{x}^i \log(1/\bar{y}^i) - x^i \log(1/y^i)\right) &\leq \sum_{i \in H} \left(\bar{x}^i \log(\frac{t+1}{t}(1/y^i)) - x_i \log(1/y^i)\right) \\
&= \sum_{i \in H} \left(\bar{x}^i \log(1/y^i) - x_i \log(1/y^i)\right) + \bar{x}^i \log(\frac{t+1}{t})) \\
&= \sum_{i \in H} \left[(\bar{x}^i - x^i) \log(1/y^i)\right] + \bar{x}_H \log(\frac{t+1}{t}) \\
&\leq \sum_{i \in H} \max\{(\bar{x}^i - x^i), 0\} \log(1/y^i) + \bar{x}_H \log(\frac{t+1}{t}) \\
&\leq \sum_{i \in H} \max\{(\bar{x}^i - x^i), 0\} \log(Nt) + \bar{x}_H \log(\frac{t+1}{t})
\end{aligned}
$$

The second inequality is by ignoring the term $i \in H$ with $\bar{x}^i < x_i$. Combining two bounds on $L, H$ gives

$$\sum_{i \in \mathcal{X}} \left(\bar{x}^i \log(1/\bar{y}^i) - x_i \log(1/y^i)\right) \leq \sum_{i \in \mathcal{X}} \max\{(\bar{x}^i - x^i), 0\} \log(Nt) + \log(\frac{t+1}{t})$$

Since there are at most one index $i \in [N]$ such that $\bar{x}^i - x^i \geq 0$ (the index of newly added data entry), and for that term we have $\bar{x}^i - x^i = \frac{1}{t+1} + \frac{t}{t+1}x^i - x^i \leq \frac{1}{t+1}$, we have

$$\sum_{i \in \mathcal{X}} \left(\bar{x}^i \log(1/\bar{y}^i) - x_i \log(1/y^i)\right) \leq \frac{1}{t+1} \log(Nt) + \log(\frac{t+1}{t})$$

$\square$

**Theorem 11** (PMWG Privacy). PMWG$(X, F, \epsilon, 0, \alpha, n)$ is $(\epsilon, 0)$-DP for $\xi$ as defined by the algorithm and

$$\epsilon = \left(1 + \frac{81}{2\alpha^2} \log N\right) \xi_n \Delta_n + \frac{81}{2\alpha^2} \sum_{t=n+1}^{\infty} \left(\frac{\log(N)}{t} + \frac{\log(t-1)}{t} + \log(\frac{t}{t-1})\right) \xi_t \Delta_t \quad \text{(B.1)}$$

*Proof.* Performing uniform update and MW update does not use any information about $X$, and therefore does not leak any privacy. Hence, the only privacy leaked is by NSG. By Theorem 24, NSG is $\epsilon$-DP for

$$\epsilon = \xi_n \Delta_n + \frac{9}{8} \sum_{t=n}^{\infty} h_t \xi_t \Delta_t$$

For convenience, let $b_n = \log N$. Because the algorithm checks for the hard query budget so that $\sum_{\tau \leq t} h_t \leq \frac{36}{\alpha^2} \sum_{\tau \leq t} b_t$ (which is motivated by Corollary 7), and that $\xi_t \Delta_t$ is non-increasing, we may upper bound $\sum_{t=n}^{\infty} h_t \xi_t \Delta_t$ by setting $h_t = \frac{36}{\alpha^2} b_t$. Hence,

$$\epsilon \leq \left(1 + \frac{81}{2\alpha^2} \log N\right) \xi_n \Delta_n + \frac{81}{2\alpha^2} \sum_{t=n+1}^{\infty} \left(\frac{\log(N)}{t} + \frac{\log(t-1)}{t} + \log(\frac{t}{t-1})\right) \xi_t \Delta_t$$

$\square$

Next we show the accuracy of PMWG.

**Theorem 12** (PMWG Accuracy). Let $k : \{n, n+1, \ldots\} \to \mathbb{R}$. On query stream $F$ such that $\sum_{\tau=n}^{t} \ell_\tau \leq \sum_{\tau=n}^{t} k_\tau$, PMWG$(X, F, \epsilon, 0, \alpha, n)$ returns an answer $a_{t,i}$ such that $|f_{t,i}(D_t) - a_{t,i}| \leq \alpha$ except with probability

$$\beta \leq \exp(-\frac{\alpha \xi_n}{24}) + 3 \sum_{t \geq n} k_t \exp(-\frac{\alpha \xi_t}{24}) \tag{B.2}$$

*Proof.* By the exact same proof in [DR14]: PMWG's $\alpha$-accuracy follows if NSG returns answers that are $\alpha/3$-accurate. Hence, we can take the $\beta$ from the NSG accuracy analysis in Theorem 26. $\square$

Finally, we describe how we choose our noise scaling function $\xi$ such that PMWG is private and accurate. That is, both $\epsilon$ and $\beta$ in lines (B.1) and (B.2) of PMWG converges.

**Observation 13.** The dominating term in the hard query bound $b_t$ in the algorithm is $\log(t-1)/t$. For (B.1) to converge, this means that we want

$$\sum_{t \geq n+1} \frac{\log(t-1)\xi_t \Delta_t}{t} = \sum_{t \geq n+1} \frac{\log(t-1)\xi_t}{t^2} \leq \int_{t=n}^{\infty} \frac{\log(t-1)\xi_t}{t^2} dt$$

to converge, so we may pick $\xi = O(t^{1-c})$ for any $c > 0$. For (B.2) to converge, we want

$$\sum_{t \geq n} k_t \exp(-\frac{\alpha \xi_t}{24})$$

to converge exponentially quickly in $\alpha$ to get a bound logarithmic in number of queries $\ell_t$. For example, we may pick $k_t = O(\exp(\frac{\alpha \xi_t}{48}))$ and $\xi_t = \Omega(t^c)$ for any $c > 0$.

By the observation, we pick $\xi_t = ct^{1/2}$ and $k_t = \kappa \exp(\frac{\alpha c t^{1/2}}{48})$ for some constant $c, \kappa$, and state our main result with this choice. Of course, other choices of $\xi_t$ and $k_t$ are possible. We will discuss other choices of $\xi_t$ and $k_t$ and their consequences after proving the following main result.

**Theorem 5.** The algorithm PMWG$(X, F, \epsilon, 0, \alpha, n)$ is $(\epsilon, 0)$-differentially private, and for any time-independent $\kappa \geq 1$ and $\beta > 0$ it is $(\alpha, \beta)$-accurate for any query stream $F$ such that $\sum_{\tau=n}^{t} \ell_\tau \leq \kappa \sum_{\tau=n}^{t} \exp(\frac{\alpha^3 \epsilon \sqrt{n\tau}}{C \log(Nn)})$ for all $t \geq n$ and sufficiently large constant $C$ as long as $N \geq 3, n \geq 21$ and

$$\alpha \geq C \left(\frac{\log(Nn) \log(\kappa n/\beta)}{n\epsilon}\right)^{1/3}.$$

We prove this without suppressing constants in the query budget and the $\alpha$ bound, i.e., we prove that $\sum_{\tau=n}^{t} \ell_\tau \leq \kappa \sum_{\tau=n}^{t} \exp(\frac{\alpha^3 \epsilon \sqrt{n\tau}}{8262 \log(Nn)})$ and $\alpha \geq (\frac{8262 \log(Nn) \log(192\kappa n/\beta)}{n\epsilon})^{1/3}$ suffice for accuracy. Note that with this choice of $\alpha$, the query budget is

$$\kappa \sum_{\tau=n}^{t} \exp\left(\frac{\alpha^3 \epsilon \sqrt{n\tau}}{8262 \log(Nn)}\right) \geq \kappa \sum_{\tau=n}^{t} \left(\frac{192\kappa n}{\beta}\right)^{\frac{17}{16}\sqrt{\frac{\tau}{n}}}$$

*Proof.* Our main result is an instantiation of the more general results in Theorems 11 and 12. In what follows, let $c$ denote the time-independent quantity such that $\xi(t) = ct^{1/2}$. Applying Theorem 11, the privacy loss of PMWG is

$$\epsilon' = \left(1 + \frac{81}{2\alpha^2}\log N\right)cn^{-1/2} + \frac{81}{2\alpha^2}\sum_{t=n+1}^{\infty}\left(\frac{\log(N)}{t} + \frac{\log(t-1)}{t} + \log(\frac{t}{t-1})\right)ct^{-1/2}$$

$$\leq \left(\frac{81}{\alpha^2}\log N\right)cn^{-1/2} + \frac{81c}{2\alpha^2}\int_{t=n}^{\infty}\left(\frac{\log(N)}{t^{3/2}} + \frac{\log(t)}{t^{3/2}} + \frac{1}{(t-1)^{3/2}}\right)dt$$

$$= \frac{81c\log N}{\alpha^2 n^{1/2}} + \frac{81c}{2\alpha^2}\left[-\frac{2\log N}{t^{1/2}} - \frac{4+2\log t}{t^{1/2}} - \frac{2}{(t-1)^{1/2}}\right]_{t=n}^{\infty}$$

$$= \frac{81c}{\alpha^2}\left(\frac{2\log N}{n^{1/2}} + \frac{2+\log n}{n^{1/2}} + \frac{1}{(n-1)^{1/2}}\right)$$

$$\leq \frac{81c}{\alpha^2}\cdot\frac{2(\log N + \log n)}{n^{1/2}}$$

where approximate the sum by integral and use $\log(t/(t-1))t^{-1/2} = \log(1 + \frac{1}{t-1})t^{-1/2} \leq \frac{1}{t-1}(t-1)^{-1/2}$ for the first inequality. The last inequality is true for $n \geq 21$. Finally, setting $c = \frac{\alpha^2 n^{1/2}}{162\log(Nn)}\epsilon$ gives $\epsilon' = \epsilon$.

Applying Theorem 12, PMWG is $(\alpha, \beta')$-accurate for

$$\beta' = \exp(-\frac{c\alpha n^{1/2}}{24}) + 3\kappa\sum_{t\geq n}\exp(\frac{c\alpha t^{1/2}}{48})\exp(-\frac{c\alpha t^{1/2}}{24})$$

$$\leq \exp(-\frac{c\alpha n^{1/2}}{24}) + 3\kappa\int_{t=n-1}^{\infty}\exp(-\frac{c\alpha t^{1/2}}{48})dt$$

$$= \exp(-\frac{c\alpha n^{1/2}}{24}) + 3\kappa\left[-\frac{96\left(\alpha c\sqrt{t} + 48\right)\exp(-\frac{\alpha c\sqrt{t}}{48})}{\alpha^2 c^2}\right]_{t=n-1}^{\infty}$$

$$= \exp(-\frac{c\alpha n^{1/2}}{24}) + 288\kappa\frac{\left(\alpha c\sqrt{n-1} + 48\right)}{\alpha^2 c^2}\exp(-\frac{\alpha c\sqrt{n-1}}{48})$$

To get $\beta' \leq \beta$, we can require $\exp(-\frac{c\alpha n^{1/2}}{24}) \leq \beta/2$ and $288\kappa\frac{\left(\alpha c\sqrt{n-1}+48\right)}{\alpha^2 c^2}\exp(-\frac{\alpha c\sqrt{n-1}}{48}) \leq \beta/2$. The first is equivalent to

$$\frac{c\alpha n^{1/2}}{24} \geq \log(2/\beta) \iff c\alpha \geq \frac{24\log(2/\beta)}{n^{1/2}} \iff \alpha \geq \left(\frac{3888\log(Nn)\log(2/\beta)}{n\epsilon}\right)^{1/3} \quad \text{(B.3)}$$

We assume that $\alpha$ satisfies (B.3) before proceeding. Secondly, (B.3) gives $c\alpha \geq \frac{24\log(2/\beta)}{n^{1/2}}$, which implies

$$\frac{\left(\alpha c\sqrt{n-1}+48\right)}{\alpha^2 c^2} \leq \frac{n^{1/2}\sqrt{n-1}}{24\log(2/\beta)} + \frac{n}{12\log^2(2/\beta)} \leq \frac{n(\log(2/\beta)+2)}{24\log^2(2/\beta)} \leq \frac{n(2+\log 2)}{24\log^2(2)} < \frac{n}{3}$$

Hence, it's enough to require $96\kappa n\exp(-\frac{\alpha c\sqrt{n-1}}{48}) \leq \beta/2$. This is equivalent to

$$\frac{\alpha c\sqrt{n-1}}{48} \geq \log(192\kappa n/\beta)$$

For $n \geq 9$, $\frac{\sqrt{n-1}}{48} \geq \frac{n^{1/2}}{51}$, so we only need

$$\frac{\alpha c}{51} \geq \frac{\log(192\kappa n/\beta)}{n^{1/2}} \iff \alpha \geq \left(\frac{8262\log(Nn)\log(192\kappa n/\beta)}{n\epsilon}\right)^{1/3} \quad \text{(B.4)}$$

(B.4) is a stronger bound than (B.3) for $\kappa \geq 1$. $\qquad\square$

What if we choose different $\xi_t$ and query budget? The earlier proof shows that as long as we have $\xi_t \Delta_t = O(t^{-p})$ for $p > 0$, the privacy loss will still converge. We state a lemma for such case here and the proof for completeness.

**Lemma 14.** Let $\xi_t \Delta_t = ct^{-q}$ for some constant $c$ independent of $t$ and $1 \geq q > 0$. Then, for $n \geq 5$, $N \geq 3$,

$$\epsilon' := \left(1 + \frac{81}{2\alpha^2}\log N\right)\xi_n\Delta_n + \frac{81}{2\alpha^2}\sum_{t=n+1}^{\infty}\left(\frac{\log(N)}{t} + \frac{\log(t-1)}{t} + \log(\frac{t}{t-1})\right)\xi_t\Delta_t$$

$$\leq \frac{126c\log(Nn)}{\alpha^2 q^2 n^q}$$

*Proof.*

$$\epsilon' = \left(1 + \frac{81}{2\alpha^2}\log N\right)cn^{-q} + \frac{81}{2\alpha^2}\sum_{t=n+1}^{\infty}\left(\frac{\log(N)}{t} + \frac{\log(t-1)}{t} + \log(\frac{t}{t-1})\right)ct^{-q}$$

$$\leq \left(\frac{81}{\alpha^2}\log N\right)cn^{-q} + \frac{81c}{2\alpha^2}\int_{t=n}^{\infty}\left(t^{-q-1}\log(N) + t^{-q-1}\log(t) + (t-1)^{-q-1}\right)dt$$

$$= \frac{81c\log N}{\alpha^2}n^{-q} + \frac{81c}{2\alpha^2}\left[-\frac{\log(N)t^{-q} + \log(t)t^{-q} + (t-1)^{-q}}{q} - \frac{t^{-q}}{q^2}\right]_{t=n}^{\infty}$$

$$\leq \frac{81c\log N}{\alpha^2}n^{-q} + \frac{81c}{2\alpha^2}\left(\frac{\log(N)n^{-q} + \log(n)n^{-q} + 2n^{-q}}{q} + \frac{n^{-q}}{q^2}\right)$$

$$\leq \frac{81c\log N}{\alpha^2}n^{-q} + \frac{81c}{2\alpha^2}\left(\frac{\log(N) + \log(n) + 3}{q^2}\right)n^{-q}$$

$$\leq \frac{81cn^{-q}}{2\alpha^2}\left(2\log N + \frac{\log(N) + \log(n) + 3}{q^2}\right)$$

$$\leq \frac{81cn^{-q}}{2\alpha^2}\left(\frac{3\log(Nn)}{q^2}\right) \leq \frac{126c\log(Nn)}{\alpha^2 q^2 n^q}$$

where we use the fact that $(n-1)^{-q} \leq 2n^{-q}$ and $\log N \leq \frac{\log N}{q^2}$ for $n \geq 2, 0 \leq q \leq 1$, and that $3 \leq 2\log n$ for $n \geq 5$. $\quad\square$

We also know that for $\xi_t = \Omega(t^p)$ for some $p > 0$, $\int_{t=n-1}^{\infty}\exp(-\frac{c\alpha\xi_t}{48})dt = P_{p,c\alpha}(n-1)\exp(-\frac{c\alpha\xi_{n-1}}{48})$ for some polynomial $P_{p,c\alpha}(n)$ dependent on $p, c\alpha$. Stating the exact bound for this integral, however, involve approximating an upper incomplete gamma function. To keep the mathematical rigor, we restrict to the case $p \geq 1/4$. Smaller $p$ involves better optimization of constants in the proof.

**Lemma 15.** Let $1 \geq p \geq 1/4, n \geq 17$ and $c, \alpha$ be constants independent of $t$ such that $c\alpha n^p \geq 24\log(2/\beta)$ and $\beta < 2^{-15/2}$. Then

$$\int_{t=n-1}^{\infty}\exp\left(-\frac{c\alpha t^p}{48}\right)dt \leq \frac{6ne^{-\frac{c\alpha n^p}{51}}}{p}$$

*Proof.* We have

$$\int_{t=n-1}^{\infty}\exp(-\frac{c\alpha t^p}{48})dt = \frac{\Gamma\left(\frac{1}{p}, \frac{c\alpha}{48}(n-1)^p\right)}{p(\frac{c\alpha}{48})^{\frac{1}{p}}}$$

where $\Gamma(s, x) = \int_x^{\infty}t^{s-1}e^{-t}dt$ is an upper incomplete gamma function. We now use the bound in [Jam16][4] that for any $a > 1, e^x > 2^a$, we have $\Gamma(a, x) \leq 2^a x^{a-1}e^{-x}$. With $n \geq 17$, $\frac{c\alpha}{48}(n-1)^p \geq$

$\frac{c\alpha n^p}{51} \geq \frac{8\log(2/\beta)}{17}$. Choosing $\beta < 2^{-15/2}$ gives $\exp(\frac{8\log(2/\beta)}{17}) \geq 2^4 \geq 2^{1/p}$, so we can apply the bound

$$\Gamma\left(\frac{1}{p}, \frac{c\alpha}{48}(n-1)^p\right) \leq \Gamma\left(\frac{1}{p}, \frac{c\alpha n^p}{51}\right) \leq 2^{1/p}\left(\frac{c\alpha n^p}{51}\right)^{\frac{1}{p}-1} e^{-\frac{c\alpha n^p}{51}}$$

Therefore,

$$\frac{\Gamma\left(\frac{1}{p}, \frac{c\alpha}{48}(n-1)^p\right)}{p(\frac{c\alpha}{48})^{\frac{1}{p}}} \leq \frac{51(32/17)^{1/p}ne^{-\frac{c\alpha n^p}{51}}}{pc\alpha n^p} \leq \frac{17(32/17)^{1/p}ne^{-\frac{c\alpha n^p}{51}}}{8p\log(2/\beta)}$$

$$\leq \frac{6ne^{-\frac{c\alpha n^p}{51}}}{p}$$

where we use $p \geq 1/4$ and $\beta < 2^{-15/2}$ to get the last inequality. $\qquad\square$

Finally, we state a generalized result in a class of noise function $\xi$ and corresponding query budget. Here we modify $\text{PMWG}(X, F, \epsilon, 0, \alpha, n, p)$ to include another parameter $p$. The modified version only changes the definition of noise function $\xi$ to be

$$\xi_t = \begin{cases} \frac{\alpha^2(1-p)^2 n^{1-p}}{126\log(Nn)}\epsilon t^p & \text{if } \delta = 0 \\ \frac{\alpha(1-p)n^{1-p}}{24\log^{1/2}(Nn)\log^{1/2}(1/\delta)}\epsilon t^p & \text{if } \delta > 0 \end{cases} \tag{B.5}$$

**Theorem 16** (Generalized-Noise PMWG $\epsilon$-DP Result). *Let $p \in [1/4, 1)$. The algorithm $\text{PMWG}(X, F, \epsilon, 0, \alpha, n, p)$ is $(\epsilon, 0)$-differentially private, and for any time-independent $\kappa \geq 1$ and $\beta \in (0, 2^{-15/2})$ it is $(\alpha, \beta)$-accurate for any query stream $F$ such that $\sum_{\tau=n}^{t} \ell_\tau \leq \kappa \sum_{\tau=n}^{t} \exp(\frac{\alpha^3(1-p)^2 \epsilon n^{1-p}\tau^p}{6048\log(Nn)})$ for all $t \geq n$ as long as $N \geq 3, n \geq 17$ and*

$$\alpha \geq \left(\frac{6426\log(Nn)\log(144\kappa n/\beta)}{(1-p)^2 n\epsilon}\right)^{1/3} \tag{B.6}$$

*Proof.* Applying Theorem 11 and Lemma 14, the privacy loss of PMWG is

$$\epsilon' \leq \frac{126c\log(Nn)}{\alpha^2 q^2 n^q}$$

where $q = 1 - p > 0$. Setting $c = \frac{\alpha^2(1-p)^2 n^{1-p}}{126\log(Nn)}\epsilon$ gives $\epsilon' = \epsilon$.

Applying Theorem 12, PMWG is $(\alpha, \beta')$-accurate for

$$\beta' \leq \exp(-\frac{c\alpha n^p}{24}) + 3\kappa \int_{t=n-1}^{\infty} \exp(-\frac{c\alpha t^p}{48})dt$$

Again, we require the first term to be at most $\beta/2$:

$$\frac{c\alpha n^p}{24} \geq \log(2/\beta) \iff \frac{(1-p)^2\alpha^3 n}{126\log(Nn)}\epsilon \geq 24\log(2/\beta) \iff \alpha \geq \left(\frac{3024\log(Nn)\log(2/\beta)}{(1-p)^2 n\epsilon}\right)^{1/3} \tag{B.7}$$

We assume that $\alpha$ satisfies this requirement before proceeding. Now we apply Lemma 15:

$$\int_{t=n-1}^{\infty} \exp(-\frac{c\alpha t^p}{48})dt \leq \frac{6ne^{-\frac{c\alpha n^p}{51}}}{p} \leq 24ne^{-\frac{c\alpha n^p}{51}}$$

So it's enough to require $3\kappa \cdot 24ne^{-\frac{c\alpha n^p}{51}} \le \beta/2$. This is equivalent to

$$\frac{c\alpha n^p}{51} \ge \log\left(\frac{144\kappa n}{\beta}\right) \iff \frac{(1-p)^2\alpha^3 n}{126\log(Nn)}\epsilon \ge 51\log\left(\frac{144\kappa n}{\beta}\right) \tag{B.8}$$

$$\iff \alpha \ge \left(\frac{6426\log(Nn)\log(144\kappa n/\beta)}{(1-p)^2 n\epsilon}\right)^{1/3} \tag{B.9}$$

Clearly the second requirement is stronger than the first for $\kappa \ge 1$. $\qquad\square$

**Observation 17.** Note that with the choice of $\alpha$ from Theorem 16, we have the query budget

$$\kappa\sum_{\tau=n}^{t}\exp\left(\frac{\alpha^3(1-p)^2\epsilon n^{1-p}\tau^p}{6048\log(Nn)}\right) \ge \kappa\sum_{\tau=n}^{t}\left(\frac{144\kappa n}{\beta}\right)^{\frac{17}{16}\left(\frac{\tau}{n}\right)^p} \tag{B.10}$$

These bounds tell us that as $p$ approaches 1, the query budget approaches exponential in $t$, but accuracy suffers proportionally to $(1-p)^{-2/3}$. The accuracy bound (B.6) is comparable to the static PMW ([HR10]), which is

$$\alpha_{\text{static}} = \Theta\left(\frac{\log N\log(k/\beta)}{n\epsilon}\right)^{1/3}.$$

Therefore, we suffer only a constant loss in accuracy as long as $n$ is bounded polynomially in $N$ and $k/\beta$. Note that, however, our query budget in the growing setting allows a generous additional number of queries to be asked upon each arrival of new data entry.

A similar observation can be made for $(\epsilon, \delta)$-DP. Theorem 19 in Section B.2 (PMWG result for $(\epsilon, \delta)$-DP) tell us that as we increase $p$ closer to 1, we increase the query budget (lower bounded by (B.10) as well) and suffer the accuracy loss proportional to $(1-p)^{-1/2}$. Similarly, the accuracy bound (B.11) in Theorem 19 for $(\epsilon, \delta)$-DP show that we suffer only a constant loss in accuracy as long as $n$ is bounded polynomially in $N$ and $k/\beta$ compared to the accuracy of the static PMW ([HR10]), which is

$$\alpha_{\text{static}} = \Theta\left(\frac{\log^{1/2}N\log(k/\beta)\log(1/\delta)}{\epsilon n}\right)^{1/2}.$$

## B.2   $(\epsilon, \delta)$-DP for PMWG

With Theorem 4, the total privacy loss from compositions may come from the *sum of square* of privacy losses, rather than the sum. We mimic the proof for $\epsilon$-DP, except that the sum is now on the square of privacy losses of all background algorithms (ATG combined with Laplace mechanism). It is straight forward to compute the sum of squares of all privacy loss of NATG, of NSG, and then of PMWG. That sum for PMWG is:

$$\tau := \left(1 + \frac{585}{16\alpha^2}\log N\right)(\xi_n\Delta_n)^2 + \frac{585}{16\alpha^2}\sum_{t=n+1}^{\infty}\left(\frac{\log(N)}{t} + \frac{\log(t-1)}{t} + \log\left(\frac{t}{t-1}\right)\right)(\xi_t\Delta_t)^2$$

We now state how to upper bound this sum, which is in a similar form to Lemma 14.

**Lemma 18.** Let $\xi_t\Delta_t = ct^{-q}$ for some constant $c$ independent of $t$ and $1 \ge q > 0$. Then, for $n \ge 3$, $N \ge 3$, if

$$\tau := \left(1 + \frac{585}{16\alpha^2}\log N\right)(\xi_n\Delta_n)^2 + \frac{585}{16\alpha^2}\sum_{t=n+1}^{\infty}\left(\frac{\log(N)}{t} + \frac{\log(t-1)}{t} + \log\left(\frac{t}{t-1}\right)\right)(\xi_t\Delta_t)^2$$

Then,

$$\sqrt{\tau} \le \frac{12cn^{-q}\log^{1/2}(Nn)}{\alpha q}$$

*Proof.* By a similar calculation as in Lemma 14 but with $c^2 n^{-2q}$ in place of $cn^{-q}$,

$$
\begin{aligned}
\tau &\leq \frac{585c^2 \log N}{8\alpha^2} n^{-2q} + \frac{585c^2}{16\alpha^2} \left( \frac{\log(N)n^{-2q} + \log(n)n^{-2q} + 2n^{-2q}}{2q} + \frac{n^{-2q}}{4q^2} \right) \\
&\leq \frac{585^2 \log N}{8\alpha^2} n^{-2q} + \frac{585c^2}{16\alpha^2} \left( \frac{2\log(N) + 2\log(n) + 5}{4q^2} \right) n^{-2q} \\
&\leq \frac{c^2 n^{-2q}}{\alpha^2} \left( \frac{585}{8} \cdot \frac{\log N}{q^2} + \frac{585}{64} \cdot \frac{2\log(N) + 10\log(n)}{q^2} \right) \\
&= \frac{c^2 n^{-2q}}{\alpha^2} \left( \frac{2925 \log(Nn)}{32q^2} \right)
\end{aligned}
$$

where we use the bounds $5 \leq 8\log(n)$ and $\log N \leq \frac{\log N}{q^2}$. The result now follows. $\qquad\square$

With the bound by Lemma 18 and Theorem 4, we achieve a $(\epsilon, \delta)$-DP version of SPMW result.

**Theorem 19** (Generalized-Noise PMWG $(\epsilon, \delta)$-DP Result)**.** Let $p \in [1/4, 1), \delta \in (0, e^{-1})$. The algorithm PMWG$(X, F, \epsilon, \delta, \alpha, n, p)$ is $(\epsilon, \delta)$-differentially private, and for any time-independent $\kappa \geq 1$ and $\beta \in (0, 2^{-15/2})$ it is $(\alpha, \beta)$-accurate for any query stream $F$ such that $\sum_{\tau=n}^{t} \ell_\tau \leq \kappa \sum_{\tau=n}^{t} \exp(\frac{\alpha^2(1-p)\epsilon n^{1-p}\tau^p}{1152 \log^{1/2}(Nn) \log^{1/2}(1/\delta)})$ for all $t \geq n$ as long as $N \geq 3, n \geq 17$ and

$$
\alpha \geq \left( \frac{1224 \log^{1/2}(Nn) \log(144\kappa n/\beta) \log^{1/2}(1/\delta)}{(1-p)n\epsilon} \right)^{1/2} \tag{B.11}
$$

Note that with this choice of $\alpha$, we have the query budget

$$
\kappa \sum_{\tau=n}^{t} \exp(\frac{\alpha^2(1-p)\epsilon n^{1-p}\tau^p}{1152 \log^{1/2}(Nn) \log^{1/2}(1/\delta)}) \geq \kappa \sum_{\tau=n}^{t} \left( \frac{144\kappa n}{\beta} \right)^{\frac{17}{16}\left(\frac{\tau}{n}\right)^p} \tag{B.12}
$$

*Proof.* Applying Lemma 18 and Theorem 4, PMWG is $(\epsilon, \delta)$-DP for

$$
\epsilon' \leq \frac{24cn^{-q} \log^{1/2}(Nn) \log^{1/2}(1/\delta)}{\alpha q}
$$

where $q = 1 - p > 0$. Setting $c = \frac{\alpha(1-p)n^{1-p}}{24\log^{1/2}(Nn)\log^{1/2}(1/\delta)}\epsilon$ gives $\epsilon' = \epsilon$. The rest of the proof now follows similarly exactly in same way as in Theorem 16, except that now $\alpha c n^p = \frac{\alpha^2(1-p)n\epsilon}{24\log^{1/2}(Nn)\log^{1/2}(1/\delta)}$ instead of $\alpha c n^p = \frac{\alpha^3(1-p)^2 n\epsilon}{126\log(Nn)}$. $\qquad\square$

Theorem 8 instantiates this previous result with $p = 1/2$.

# C  Sparse Vector Algorithms for Growing Databases

In this section, we give descriptions of three primitive algorithms applying the sparse vector technique modified for the dynamic setting of growing databases, which we call above threshold for growing databases (ATG), numeric above threshold for growing databases (NATG), and numeric sparse for growing databases (NSG). The main difference in analyzing privacy and accuracy for dynamic algorithms is that the results now depend on a changing database size. In this section, the database has size $t_0$ and the initialization time $t_0$ and it grows by one entry each time step. This is why results are usually stated in terms dependent on time such as $t_0, t$.

## C.1  Above Threshold for Growing Databases

Before analyzing numeric above threshold for growing databases (NATG, Algorithm C.2), we consider the simpler above threshold algorithm (ATG). This algorithm is simply NATG when $\top$ is output for above threshold queries rather than noisy numeric answers.

**Theorem 20** (Privacy of ATG). *Let* $D = \{D_{t_0}, D_{t_0+1} \ldots\}, D' = \{D'_{t_0}, D'_{t_0+1} \ldots\}$ *be two sequences of databases of size* $t_0, t_0+1, \ldots$ *such that for each* $t, D_t \sim D'_t$. *Let* $\xi, \Delta : \{t_0, t_0+1, \ldots\} \to \mathbb{R}^+$ *be such that both* $\Delta, \xi \cdot \Delta$ *are non-increasing and* $\xi$ *is non-decreasing. Let* $F = \{\{f_{t,j}\}_{j=1}^{\ell_t}\}_{t \geq n}$ *be the stream of queries such that the sensitivity* $\Delta_{f_{t,j}} \leq \Delta_t$ *for all* $t, j$. *Then for all possible output* $a$ *by ATG,*

$$\Pr\left[ATG(D, F, T, \xi) = a\right] \leq \exp(\xi_{t_0} \Delta_{t_0}) \Pr\left[ATG(D', F, T, \xi) = a\right]$$

*Proof.* We follow the proof as in [DR14], but modify it slightly. Let $A, A'$ represent the random variables of output of ATG running on $D, D'$, respectively. Suppose ATG halts at the last query $f_{t',i'}$. Let $a$ denote this output, i.e. $a_{t,i} = \bot$ for all $(t, i) < (t', i')$ (i.e. indices of all queries that comes before $(t', i')$) and $a_{t',i'} = \top$. Define

$$H(D) = \max_{(t,i) < (t',i')} \xi_t \cdot (f_{t,i}(D_t) + \nu_{t,i} - T)$$

Fix $\nu_{t,i}$ for all $(t, i) < (t', i')$, so that $H(D)$ is a deterministic quantity. Then,

$$
\begin{aligned}
\Pr_{\eta, \nu_{t',i'}} [A = a] &= \Pr_{\eta, \nu_{t',i'}} \left[ f_{t,i}(D_t) + \nu_{t,i} < \hat{T}_t, \ \forall (t,i) < (t',i') \text{ and } f_{t',i'}(D_{t'}) + \nu_{t',i'} \geq \hat{T}_{t'} \right] \\
&= \Pr_{\eta, \nu_{t',i'}} \left[ H(D) < \eta \text{ and } \xi_{t'} \cdot (f_{t',i'}(D_{t'}) + \nu_{t',i'} - T) \geq \eta \right] \\
&= \Pr_{\eta, \nu_{t',i'}} \left[ \eta \in (H(D), \xi_{t'} \cdot (f_{t',i'}(D_{t'}) + \nu_{t',i'} - T)] \right] \\
&= \int_{-\infty}^{\infty} \int_{-\infty}^{\infty} \Pr_{\nu_{t',i'}} [\nu_{t',i'} = v] \\
&\qquad \cdot \Pr_{\eta} [\eta = \eta_0] \mathbb{1}[\eta_0 \in (H(D), \xi_{t'} \cdot (f_{t',i'}(D_{t'}) + v - T)] \, dv \, d\eta_0 \\
&:= *
\end{aligned}
$$

Change the variable as follows:

$$
\begin{aligned}
\hat{v} &= v + \frac{H(D) - H(D')}{\xi_{t'}} + f_{t',i'}(D_{t'}) - f_{t',i'}(D'_{t'}) \\
\hat{\eta}_0 &= \eta_0 + H(D) - H(D')
\end{aligned}
$$

We have $|H(D) - H(D')| \leq \max_{t \geq t_0} \xi_t \Delta_t = \xi_{t_0} \Delta_{t_0}$ (by $\xi_t \Delta_t$ being non-increasing), and $|f_{t',i'}(D_{t'}) - f_{t',i'}(D'_{t'})| \leq \Delta_{t'}$. Therefore,

$$|\hat{v} - v| \leq \frac{\xi_{t_0} \Delta_{t_0}}{\xi_{t'}} + \Delta_{t'}, |\hat{\eta}_0 - \eta_0| \leq \xi_{t_0} \Delta_{t_0}$$

Apply this change of variable to get

$$* = \int_{-\infty}^{\infty} \int_{-\infty}^{\infty} \Pr_{\nu_{t',i'}} \left[ \nu_{t',i'} = \hat{v} \right] \Pr_{\eta} \left[ \eta = \hat{\eta}_0 \right] \mathbb{1}[\eta_0 + H(D) - H(D') \in$$

$$(H(D), \xi_{t'} \cdot (f_{t',i'}(D_{t'}) + \hat{v} - T)] \, dv \, d\eta_0$$

$$= \int_{-\infty}^{\infty} \int_{-\infty}^{\infty} \Pr_{\nu_{t',i'}} \left[ \nu_{t',i'} = \hat{v} \right] \Pr_{\eta} \left[ \eta = \hat{\eta}_0 \right] \mathbb{1}[\eta_0 \in$$

$$(H(D'), \xi_{t'} \cdot (f_{t',i'}(D_{t'}) + \hat{v} - T) + H(D') - H(D)] \, dv \, d\eta_0$$

$$= \int_{-\infty}^{\infty} \int_{-\infty}^{\infty} \Pr_{\nu_{t',i'}} \left[ \nu_{t',i'} = \hat{v} \right] \Pr_{\eta} \left[ \eta = \hat{\eta}_0 \right] \mathbb{1}[\eta_0 \in$$

$$(H(D'), \xi_{t'} \cdot (f_{t',i'}(D'_{t'}) + v - T)] \, dv \, d\eta_0$$

$$\leq \int_{-\infty}^{\infty} \int_{-\infty}^{\infty} \exp\left( \frac{\xi_{t'}}{4} \left( \frac{\xi_{t_0} \Delta_{t_0}}{\xi_{t'}} + \Delta_{t'} \right) \right) \Pr_{\nu_{t',i'}} \left[ \nu_{t',i'} = v \right] \exp\left( \frac{\xi_{t_0} \Delta_{t_0}}{2} \right) \Pr_{\eta} \left[ \eta = \eta_0 \right]$$

$$\cdot \mathbb{1}[\eta_0 \in (H(D'), \xi_{t'} \cdot (f_{t',i'}(D'_{t'}) + v - T)] \, dv \, d\eta_0$$

$$= \exp\left( \frac{\xi_{t'}}{4} \left( \frac{\xi_{t_0} \Delta_{t_0}}{\xi_{t'}} + \Delta_{t'} \right) + \frac{\xi_{t_0} \Delta_{t_0}}{2} \right)$$

$$\cdot \Pr_{\eta, \nu_{t',i'}} \left[ H(D') < \eta \text{ and } \xi_{t'} \cdot (f_{t',i'}(D'_{t'}) + \nu_{t',i'} - T) \geq \eta \right]$$

$$= \exp\left( \frac{\xi_{t_0} \Delta_{t_0}}{4} + \frac{\xi_{t'} \Delta_{t'}}{4} + \frac{\xi_{t_0} \Delta_{t_0}}{2} \right) \Pr_{\eta, \nu_{t',i'}} \left[ A' = a \right]$$

$$\leq \exp\left( \xi_{t_0} \Delta_{t_0} \right) \Pr_{\eta, \nu_{t',i'}} \left[ A' = a \right]$$

The first inequality comes from the bounds on $|\hat{v} - v|, |\hat{\eta}_0 - \eta_0|$ and the pdf of Laplace distribution. The last inequality is by $\xi_t \Delta_t$ being non-increasing. □

Next, we define and prove the accuracy statement.

**Definition 5** (Accuracy of Threshold Answers). An algorithm which outputs answers $a_{t,i} \in \{\top, \bot\}^*$ to queries $F = (f_{t,i})_{(t,i) \leq (t',i')}$ over a sequence of growing database $D = \{D_{t_0}, D_{t_0+1}, \ldots\}$ is $(\alpha, \beta)$-accurate with respect to threshold $T$ and stream of queries $F$ if with probability at least $1 - \beta$, the algorithm does not halt before $(t', i')$, and that for all $a_{t,i} = \top$,

$$f_{t,i}(D_t) \geq T - \alpha$$

and for all $a_{t,i} = \bot$,

$$f_{t,i}(D_t) \leq T + \alpha$$

We define the algorithm to be $\alpha$-accurary if this event (which is true with probability at least $1 - \beta$) is satisfied.

Now we prove that ATG is accurate.

**Theorem 21** (Accurary of ATG). Use the same notation and with the same assumptions as in Theorem 20. For any stream of queries $F = (f_{t,i})_{(t,i) \leq (t',i')}$ and growing database $D = \{D_{t_0}, D_{t_0+1}, \ldots\}$ such that for all $(t, i) < (t', i')$, $f_{t,i}(D_t) < T - \alpha$ and $f_{t',i'}(D_{t'}) \geq T + \alpha$, $\text{ATG}(D, F, T, \xi)$ is $(\alpha, \beta)$-accurate with respect to threshold $T$ and stream of queries $F$ for

$$\beta = \sum_{t=t_0}^{t'} \ell_t \exp\left( -\frac{\alpha \xi_t}{8} \right) + \exp\left( -\frac{\alpha \xi_{t_0}}{8} \right)$$

*Proof.* First, we want to show that $a_{t,i} = \bot$ if and only if $(t, i) < (t', i')$ with high probability. This is true if we can show that for all $(t, i)$,

$$\left| \nu_{t,i} - \frac{\eta}{\xi_t} \right| \leq \alpha \tag{C.1}$$

Because if so, we have that for all $(t,i) < (t',i')$, $f_{t,i}(D_t) + \nu_{t,i} < (T-\alpha) + (\alpha + \frac{\eta}{\xi_t}) = \hat{T}_t$, so ATG will output $\perp$, and that $f_{t',i'}(D_{t'}) + \nu_{t',i'} \geq (T+\alpha) + (\frac{\eta}{\xi_t} - \alpha) = \hat{T}_t$, so ATG will output $a_{t',i'} = \top$. To show (C.1), it is sufficient to require $|\nu_{t,i}| \leq \alpha/2$ and $|\frac{\eta}{\xi_t}| \leq \alpha/2$ for all $(t,i)$. The first requirement is false with probability $\exp(-\frac{\alpha}{2}\frac{\xi_t}{4}) = \exp(-\frac{\alpha\xi_t}{8})$ for each $(t,i)$. The second requirement is equivalent to $|\frac{\eta}{\xi_{t_0}}| \leq \alpha/2$ for a single time step $t_0$, which is false with probability $\exp(-\frac{\alpha\xi_{t_0}}{4})$. By union bound, (C.1) is true except probability at most $\exp(-\frac{\alpha\xi_{t_0}}{4}) + \sum_{t=t_0}^{t'} \ell_t \exp(-\frac{\alpha\xi_t}{8}) \leq \exp(-\frac{\alpha\xi_{t_0}}{8}) + \sum_{t=t_0}^{t'} \ell_t \exp(-\frac{\alpha\xi_t}{8})$. $\qquad\square$

## C.2 Numeric Above Threshold for Growing Databases

Next we analyze privacy loss for NATG (Algorithm C.2).

---
**Algorithm 2** NATG$(D, F, T, \xi)$
---
  **for** each query $f_{t,i}$ **do**
    **if** $i = 1$ **then**
      $\hat{T}_t = T + \text{Lap}(\frac{2}{\xi_t})$.                                # same noisy threshold for all $f_{t,\cdot}$
    **end if**
    $\nu_{t,i} = \text{Lap}(\frac{4}{\xi_t})$.
    **if** $f_{t,i}(D_t) + \nu_{t,i} \geq \hat{T}_t$ **then**
      Output $a_{t,i} = f_{t,i}(D_t) + \text{Lap}(\frac{8}{\xi_t})$.
      Halt.
    **else**
      Output $a_{t,i} = \perp$.
    **end if**
  **end for**
---

**Theorem 22** (Privacy of NATG). Using the same notation and with the same assumptions as in Theorem 20, and let $t'$ denote the time step that NATG$(D, F, T, \xi)$ halts. then for all possible output $a$ by NATG,

$$\Pr\left[NATG(D, F, T, \xi) = a\right] \leq \exp\left(\xi_{t_0}\Delta_{t_0} + \frac{\xi_{t'}\Delta_{t'}}{8}\right) \Pr\left[NATG(D', F, T, \xi) = a\right]$$

*Proof.* NATG privacy loss is the sum of ATG privacy loss and the loss by Laplace noise $\text{Lap}(\frac{8}{\xi_{t'}})$ added to the numeric answer $a_{t',i'}$. The first is $\Delta_{t_0}\xi_{t_0}$ by Theorem 20 and the latter is $\xi_{t'}\Delta_{t'}/8$ because $f_{t',i}$ has sensitivity at most $1/\Delta_{t'}$. $\qquad\square$

Next, we define and prove the accuracy statement.

**Definition 6** (Accuracy of Threshold and Numeric Answers). An algorithm which outputs answers $a_{t,i} \in (\mathbb{R} \cup \perp)^*$ to a stream of queries $F = (f_{t,i})_{(t,i) \leq (t',i')}$ over a sequence of growing database $D = \{D_{t_0}, D_{t_0+1}, \ldots\}$ is $(\alpha, \beta)$-accurate with respect to threshold $T$ and stream of queries $F$ if with probability at least $1 - \beta$, the algorithm does not halt before $(t', i')$, and that for all $a_{t,i} \in \mathbb{R}$,

$$|f_{t,i}(D_t) - a_{t,i}| \leq \alpha \text{ and } f_{t,i}(D_t) \geq T - \alpha$$

and for all $a_{t,i} = \perp$,

$$f_{t,i}(D_t) \leq T + \alpha$$

We define the algorithm to be $\alpha$-accurary if this event (which is true with probability at least $1 - \beta$) is satisfied.

Now we prove that NATG is accurate.

**Theorem 23** (Accurary of NATG). Use the same notation and with the same assumptions as in Theorem 20. For any sequence of queries $F = (f_{t,i})_{(t,i) \leq (t',i')}$ and growing database

$D = \{D_{t_0}, D_{t_0+1}, \ldots\}$ such that for all $(t, i) < (t', i')$, $f_{t,i}(D_t) < T - \alpha$ and $f_{t',i'}(D_{t'}) \geq T + \alpha$, NATG$(D, F, T, \xi)$ is $(\alpha, \beta)$-accurate with respect to threshold $T$ and stream of queries $F$ for

$$\beta = \sum_{t=t_0}^{t'} \ell_t \exp(-\frac{\alpha \xi_t}{8}) + \exp(-\frac{\alpha \xi_{t_0}}{8}) + \exp(-\frac{\alpha \xi_{t'}}{8})$$

*Proof.* We apply Theorem 21 so that except with at most probability $\sum_{t=t_0}^{t'} \ell_t \exp(-\frac{\alpha \xi_t}{8}) + \exp(-\frac{\alpha \xi_{t_0}}{8})$, NATG is accurate for threshold answers, i.e. $f_{t,i}(D_t) \geq T - \alpha$ and $f_{t,i}(D_t) \leq T + \alpha$ hold for $a_{t,i} \in \mathbb{R}$ and $a_{t,i} = \bot$, respectively.

It's left to show that $|f_{t',i'}(D_{t'}) - a_{t',i'}| \leq \alpha$. This is true except with probability $\exp(-\frac{\alpha \xi_{t'}}{8})$ by Laplace mechanism. Therefore,

$$\beta \leq \sum_{t=t_0}^{t'} \ell_t \exp(-\frac{\alpha \xi_t}{8}) + \exp(-\frac{\alpha \xi_{t_0}}{8}) + \exp(-\frac{\alpha \xi_{t'}}{8})$$

$\square$

### C.3 Numeric Sparse for Growing Databases

We now compose NATG multiple times into numeric sparse for growing databases (NSG). Note that we may run NATG infinitely many times as long as there is input coming online. Any query that causes NATG to output a number and halt is called hard; any other query is called easy.

---

**Algorithm 3** NSG$(D, F, T, \xi)$

---

**for** Each query $f_{t,i}$ **do**
    **if** no NATG subroutine is currently running **then**
        Initialize a NATG subroutine with the same arguments
    **end if**
    Output the NATG subroutine's output for $f_{t,i}$.
**end for**

---

Analysis of privacy of NSG can be done by simply summing up the privacy loss stated for NATG. Since privacy for NATG changes over time, this privacy loss is then dependent on the time that NATG starts and ends, i.e. when hard queries come.

**Theorem 24** (Privacy of NSG). Using the same notation and with the same assumptions as in Theorem 20. Suppose that NSG starts at time $n$ with $h_t$ hard queries arriving at time $t$ for each $t \geq n$, then NSG is $(\epsilon, 0)$-DP for

$$\epsilon = \xi_n \Delta_n + \frac{9}{8} \sum_{t=n}^{\infty} h_t \xi_t \Delta_t$$

*Proof.* Let $t_0^j, t_1^j$ be the start and end time of $j$th round of NATG in NSG. Then by Theorem 22, the privacy loss is at most

$$\epsilon := \sum_j \left( \xi_{t_0^j} \Delta_{t_0^j} + \frac{\xi_{t_1^j} \Delta_{t_1^j}}{8} \right)$$

The start time $t_0^j$ of round $j$ is at least the end time of last round $t_1^{j-1}$ for each $j \geq 2$, and $t_0^1 \geq n$, so

$$\epsilon \leq \xi_n \Delta_n + \sum_j \left( \xi_{t_1^j} \Delta_{t_1^j} + \frac{\xi_{t_1^j} \Delta_{t_1^j}}{8} \right)$$

$$= \xi_n \Delta_n + \frac{9}{8} \sum_{t=n}^{\infty} h_t \xi_t \Delta_t$$

The last equality is by the fact that the end times of NATG are exactly when hard queries come.

$\square$

Now we apply Theorem 24 to a more specific setting of linear queries.

**Corollary 25.** For two neighboring sequences of growing databases $D \sim D'$ and adaptively chosen linear queries $f_{t,i}$, NSG which starts at time $n$, with noise function $\xi_t = t^p$ for some $p \in [0,1]$, is $(n^{p-1} + \frac{9}{8}\sum_{t=n}^{\infty} h_t t^{p-1}, 0)$-DP.

*Proof.* From the fact that $\xi_t$ is non-decreasing, that linear queries have sensitivity $\Delta_t = 1/t$, and that $\xi_t \Delta_t = t^{p-1}$ is non-increasing, the conditions satisfy Theorem 20's assumption, so we can apply Theorem 24. $\square$

Corollary 25 suggests that, in order to bound privacy loss, we need to bound the number of hard queries, especially those arriving early in time. These arrival times of hard queries, of course, depend on specific application of NSG.

Now we state the accuracy for NSG. Note that we use the same Definition 6 from NATG, since the output of NSG and NATG are in the same format $(\mathbb{R} \cup \{\bot\})^*$.

**Theorem 26** (Accuracy of NSG). Use the same notation and with the same assumptions as in Theorem 20. Let $k : \{t_0, t_0 + 1, \ldots\} \to \mathbb{R}$. For a growing database $D = \{D_n, D_{n+1}, \ldots\}$, let $h_t = |\{i : f_{t,i}(D_t) \geq T - \alpha\}|$. Then for any sequence of queries $F = \{\{f_{t,j}\}_{j=1}^{\ell_t}\}_{t \geq n}$ such that $\sum_{\tau=t_0}^{t} \ell_t \leq \sum_{\tau=t_0}^{t} k_t$ for all $t \geq t_0$, $\text{NSG}(D, F, T, \xi)$ is $(\alpha, \beta)$-accurate with respect to threshold $T$ and stream of queries $F$ for

$$\beta = \exp(-\frac{\alpha \xi_n}{8}) + \sum_{t=n}^{\infty} (\ell_t + 2h_t) \exp(-\frac{\alpha \xi_t}{8}) \tag{C.2}$$

$$\leq \exp(-\frac{\alpha \xi_n}{8}) + 3\sum_{t=n}^{\infty} k_t \exp(-\frac{\alpha \xi_t}{8}) \tag{C.3}$$

*Proof.* We need to show that except with probability at most $\beta$:

1. For each $a_{t,i} = \bot$, $f_{t,i}(D_t) \leq T + \alpha$

2. For each $a_{t,i} \in \mathbb{R}$, $f_{t,i}(D_t) \geq T - \alpha$

3. For each $a_{t,i} \in \mathbb{R}$, $|f_{t,i}(D_t) - a_{t,i}| \leq \alpha$

Suppose the $j$th round of NATG starts and ends at time $t_0^j, t_1^j$, and answers $\ell_t^j$ queries at time $t$. The set of three conditions is equivalent to requiring that $j$th NATG round satisfies Definition 6 with respect to threshold $T$ and stream of queries that $j$th round of NATG answers. By Theorem 23 and union bound, the rest of the proof is a calculation: we can take $\beta$ to be

$$\beta = \sum_j \left( \sum_{t=n}^{\infty} \ell_t^j \exp(-\frac{\alpha \xi_t}{8}) + \exp(-\frac{\alpha \xi_{t_0^j}}{8}) + \exp(-\frac{\alpha \xi_{t_1^j}}{8}) \right)$$

$$\leq \sum_j \left( \sum_{t=n}^{\infty} k_t^j \exp(-\frac{\alpha \xi_t}{8}) \right) + \sum_j \left( \exp(-\frac{\alpha \xi_{t_1^{j-1}}}{8}) + \exp(-\frac{\alpha \xi_{t_1^{j-1}}}{8}) \right)$$

$$\leq \left( \sum_{t=n}^{\infty} \ell_t \exp(-\frac{\alpha \xi_t}{8}) \right) + \exp(-\frac{\alpha \xi_n}{8}) + \sum_j 2\exp(-\frac{\alpha \xi_{t_1^j}}{8})$$

The first inequality is by the fact that start time of the next round is after the end of the current round, and we let $t_1^0 = n$ for convenience. By condition (2), NATG can only halt on queries $f_{t,i}$ such that $f_{t,i}(D_t) \geq T - \alpha$, so there are at most $h_t$ rounds of NATG halting at time $t$. Therefore,

$$2\sum_j \exp(-\frac{\alpha \xi_{t_1^j}}{8}) \leq 2\sum_{t=n}^{\infty} h_t \exp(-\frac{\alpha \xi_t}{8})$$

which finishes the proof for (C.2). C.3 is noting that $\ell_t \geq h_t$, the definition of query budget, and that $\exp(-\frac{\alpha \xi_t}{8})$ is a non-increasing function of $t$. $\qquad\square$

# D Analysis of Black Box Schedulers

In this appendix we present the algorithms and proofs that were omitted in Section 4. Section D.1 contains BBSCHEDULER and related proofs. Section D.2 contains BBIMPROVER and related proofs.

## D.1 Fixed Accuracy as Data Accumulate

---

**Algorithm 4** BBSCHEDULER($X, F, \mathcal{M}, \epsilon, \delta, \beta, n, p, g$)

---

**if** $\delta = 0$ **then**

    Let $\gamma \leftarrow g^{\frac{1}{2p+1}} \left( \frac{\log \frac{1}{\beta}}{\epsilon n} \right)^{\frac{p}{2p+1}}$

**else**

    Let $\gamma \leftarrow g^{\frac{1}{1.5p+1}} \left( \frac{\log \frac{1}{\beta}}{\epsilon n} \right)^{\frac{p}{1.5p+1}}$

**end if**

Let $i \leftarrow -1$

**for** $t \leftarrow n, n+1, ...$ **do**

    **if** $t = (1+\gamma)^{i+1} n$ **then**

        $i \leftarrow i + 1$

        **if** $\delta = 0$ **then**

            Let $\epsilon_i \leftarrow \frac{\gamma^2(i+1)}{(1+\gamma)^{i+2}} \epsilon$

        **else**

            Let $\epsilon_i \leftarrow \frac{\gamma^{1.5}(i+1)}{(1+\gamma)^{i+1.5}} \frac{\epsilon}{3\sqrt{\log(1/\delta)}}$

        **end if**

        Let $\beta_i \leftarrow \left( \frac{\beta}{1+\beta} \right)^{i+1}$

        Let $\alpha_i \leftarrow g \left( \frac{\log \frac{1}{\beta_i}}{\epsilon_i (1+\gamma)^i n} \right)^p$

        Let $y_i \leftarrow \mathcal{M}(x_t, \epsilon_i, \alpha_i, \beta_i)$

    **end if**

    **for** $j \leftarrow 1, ..., \ell_t$ **do**

        Output $y_i(f_{t,j})$

    **end for**

**end for**

---

**Theorem 27.** Let $\mathcal{M}$ be a $(p, g)$-black box for query class $\mathcal{F}$. Then for any database stream $X$ and stream of linear queries $F$ over $\mathcal{F}$, BBSCHEDULER($X, F, \mathcal{M}, \epsilon, \delta, \beta, n, p, g$) is $(\epsilon, \delta)$-differentially private for $\epsilon < 1$ and $(\alpha, \beta)$-accurate for sufficiently large constant $C$ and

$$
\alpha \geq \begin{cases} Cg^{\frac{1}{2p+1}} \left( \frac{\log(1/\beta)}{\epsilon n} \right)^{\frac{p}{2p+1}} & \text{if } \delta = 0 \\ Cg^{\frac{1}{1.5p+1}} \left( \frac{\sqrt{\log(1/\delta)}\log(1/\beta)}{\epsilon n} \right)^{\frac{p}{1.5p+1}} & \text{if } \delta > 0 \end{cases}.
$$

*Proof.* We begin with the privacy guarantees of BBSCHEDULER. When $\delta = 0$, BBSCHEDULER runs $\mathcal{M}$ in each epoch $i$ with privacy parameter $\epsilon_i = \frac{\gamma^2(i+1)}{(1+\gamma)^{i+2}} \epsilon$. Then by Basic Composition (Theorem 3), BBSCHEDULER is $(\sum_{i=0}^{\infty} \epsilon_i, 0)$-differentially private, where

$$
\sum_{i=0}^{\infty} \epsilon_i = \frac{\gamma^2}{1+\gamma} \epsilon \sum_{i=0}^{\infty} \frac{i+1}{(1+\gamma)^{i+1}} = \epsilon.
$$

The sum $\sum_{i=0}^{\infty} \frac{i+1}{(1+\gamma)^{i+1}}$ converges to $\frac{1+\gamma}{\gamma^2}$, so BBSCHEDULER is $(\epsilon, 0)$-differentially private.

When $\delta > 0$, BBSCHEDULER runs $\mathcal{M}$ with privacy parameter $\epsilon_i = \frac{\gamma^{1.5}(i+1)}{(1+\gamma)^{i+1.5}} \frac{\epsilon}{3\sqrt{\log(1/\delta)}}$ in each epoch $i$. By CDP composition (Theorem 4), the total privacy loss is at most $\frac{1}{2} \sum_{i=0}^{\infty} \epsilon_i^2 + \sqrt{2 \left( \sum_{i=0}^{\infty} \epsilon_i^2 \right) \log(1/\delta)}$. Note that

$$\sum_{i=0}^{\infty} \epsilon_i^2 = \frac{\epsilon^2}{9\log(1/\delta)} \frac{\gamma^3}{1+\gamma} \sum_{i=0}^{\infty} \frac{(i+1)^2}{(1+\gamma)^{2(i+1)}} = \frac{\epsilon^2}{9\log(1/\delta)} \frac{\gamma^3}{1+\gamma} \frac{(1+\gamma)^2(\gamma^2 + 2\gamma + 2)}{\gamma^3(\gamma+2)^3} \leq \frac{2\epsilon^2}{9\log(1/\delta)}$$

where we used the fact that $\gamma \in (0, 1)$. Then the total privacy loss is at most

$$\frac{\epsilon^2}{9\log(1/\delta)} + \frac{2\epsilon}{3\sqrt{\log(1/\delta)}}\sqrt{\log(1/\delta)} \leq \epsilon$$

since $\epsilon < 1$.

To prove the accuracy of BBSCHEDULER we require the following lemma, which bounds the additive error introduced by answering queries that arrive mid-epoch using the slightly outdated database from the end of the previous epoch.

**Lemma 28.** For any linear query $f$ and databases $x_t$ and $x_\tau$ from a database stream $X$, where $\tau \in [t, (1+\gamma)t]$ for some $\gamma \in (0, 1)$, we have $|x_\tau(f) - x_t(f)| \leq \frac{\gamma}{1+\gamma}$.

*Proof. (Lemma 28)* The linear query $x_t(f)$ can be written in the following form: $x_t(f) = \frac{1}{t}\sum_{i=1}^{N} tx_t^i f^i$. Then since $x_\tau(f) - x_t(f) = \frac{1}{\tau}\sum_{i=1}^{N} \tau x_\tau^i f^i - \frac{1}{t}\sum_{i=1}^{N} tx_t^i f^i$, we have,

$$x_\tau(f) - x_t(f) \leq \frac{(\tau - t) + tx_t^i}{\tau} - \frac{tx_t^i}{t} \leq \frac{(\tau - t) + tx_t^i}{\tau} - \frac{tx_t^i}{\tau} = 1 - \frac{t}{\tau},$$

$$x_\tau(f) - x_t(f) \geq \frac{tx_t^i}{\tau} - \frac{tx_t^i}{t} \geq t\left(\frac{1}{\tau} - \frac{1}{t}\right) = \frac{t}{\tau} - 1.$$

The last inequality follows because $\frac{1}{\tau} - \frac{1}{t} \leq 0$ and $tx_t^i \leq t$. Thus, $|x_\tau(f) - x_t(f)| \leq 1 - \frac{t}{\tau}$. Since $\tau \in [t, (1+\gamma)t]$, then,

$$1 - \frac{t}{\tau} \leq 1 - \frac{t}{(1+\gamma)t} = 1 - \frac{1}{1+\gamma} = \frac{\gamma}{1+\gamma}.$$

$\square$

We now continue to prove the accuracy of BBSCHEDULER. Let $t_i = (1+\gamma)^i n$. Recall that epoch $i$ is defined as the time interval where $t \in \{t_i, t_i + 1, ..., t_{i+1} - 1\}$. Let $F_i$ denote the set of all queries received during epoch $i$. All queries $f \in F_i$ will be answered using $y_i$, which is computed on database $x_{t_i}$.

We want to show that $y_i(f)$ is close to $x_t(f)$ for all $f \in F_i$. Since $y_i$ is the output of $\mathcal{M}(x_{t_i}, \epsilon_i, \alpha_i, \beta_i)$, we know that for $f \in F_i$,

$$|y_i(f) - x_{t_i}(f)| \leq \alpha_i.$$

By the triangle inequality and Lemma 28, for any $f \in F_i$,

$$|y_i(f) - x_t(f)| \leq |y_i(f) - x_{t_i}(f)| + |x_{t_i}(f) - x_t(f)|$$
$$= \alpha_i + \frac{\gamma}{1+\gamma}. \tag{D.1}$$

When $\delta = 0$, we have

$$\alpha_i = g\left(\frac{\log\frac{1}{\beta_i}}{\epsilon_i n_i}\right)^p = g\left(\frac{(i+1)\log\frac{1+\beta}{\beta}}{\frac{\gamma^2(i+1)}{(1+\gamma)^{i+2}}\epsilon(1+\gamma)^i n}\right)^p = g\left(\frac{(1+\gamma)^2}{\gamma^2 \epsilon n}\log\frac{1+\beta}{\beta}\right)^p$$

Let $Z = \frac{\log\frac{1+\beta}{\beta}}{\epsilon n}$. Note that since $\gamma < 1$, we have $(1+\gamma) \in (1, 2)$. Then (D.1) becomes

$$\boxed{(D.1)} \leq \alpha_i + \gamma \leq gZ^p\left(\frac{1}{2}\gamma\right)^{-2p} + \gamma.$$

Since we set $\gamma = g^{\frac{1}{2p+1}} Z^{\frac{p}{2p+1}}$, we have:

$$gZ^p \left(\frac{1}{2}\gamma\right)^{-2p} + \gamma = gZ^p \left(\frac{1}{2} g^{\frac{1}{2p+1}} Z^{\frac{p}{2p+1}}\right)^{-2p} + g^{\frac{1}{2p+1}} Z^{\frac{p}{2p+1}}$$

$$= C_1 g^{1 - \frac{2p}{2p+1}} Z^{p - \frac{p^2}{2p+1}} + g^{\frac{1}{2p+1}} Z^{\frac{p}{2p+1}}$$

$$= C_2 g^{\frac{1}{2p+1}} Z^{\frac{p}{2p+1}}$$

where $C_1$ and $C_2$ are positive absolute constants.

When $\delta > 0$, BBSCHEDULER uses a different setting of $\gamma$ and $\epsilon_i$. Let $Z = \frac{3\sqrt{\log(1/\delta)} \log \frac{1+\beta}{\beta}}{\epsilon n}$. In this case, we have

$$\alpha_i = g\left(\frac{\log \frac{1}{\beta_i}}{\epsilon_i n_i}\right)^p = g\left(\frac{(i+1)\log\frac{1+\beta}{\beta}}{\frac{\gamma^{1.5}(i+1)}{(1+\gamma)^{i+1.5}}\frac{\epsilon}{3\sqrt{\log(1/\delta)}}(1+\gamma)^i n}\right)^p = gZ^p \left(\frac{1+\gamma}{\gamma}\right)^{1.5p} \leq gZ^p \left(\frac{1}{2}\gamma\right)^{1.5p}$$

Since we set $\gamma = g^{\frac{1}{1.5p+1}} Z^{\frac{p}{1.5p+1}}$, we have:

$$\boxed{(D.1)} \leq gZ^p \left(\frac{1}{2} g^{\frac{1}{1.5p+1}} Z^{\frac{p}{1.5p+1}}\right)^{-1.5p} + g^{\frac{1}{1.5p+1}} Z^{\frac{p}{1.5p+1}}$$

$$\leq C_1 g^{1 - \frac{1.5p}{1.5p+1}} Z^{p - \frac{1.5p^2}{1.5p+1}} + g^{\frac{1}{1.5p+1}} Z^{\frac{p}{1.5p+1}}$$

$$\leq C_2 g^{\frac{1}{1.5p+1}} Z^{\frac{p}{1.5p+1}}$$

where $C_1$ and $C_2$ are positive absolute constants.

The final accuracy bound for any $\delta \geq 0$ follows by substitution and by noting that $\log \frac{1+\beta}{\beta} = O(\log \frac{1}{\beta})$ since $\beta \in (0,1)$. Each of the $\alpha_i$ bounds holds with probability $1 - \beta_i$, so by a union bound, all will hold simultaneously with probability $1 - \sum_{i=0}^{\infty} \beta_i$, where,

$$\sum_{i=0}^{\infty} \beta_i = \frac{\beta}{2n_i^2} \leq \sum_{t=n}^{\infty} \frac{\beta}{2t^2} \leq \frac{\pi^2}{12}\beta \leq \beta.$$

Then with probability at least $1 - \beta$, all queries are answered with accuracy $Cg^{\frac{1}{2p+1}} Z^{\frac{p}{2p+1}}$ for $\delta = 0$ and $Cg^{\frac{1}{1.5p+1}} Z^{\frac{p}{1.5p+1}}$ for $\delta > 0$ for some positive absolute constant $C$. $\qquad\square$

### D.2 Improving Accuracy as Data Accumulate

---

**Algorithm 5** BBIMPROVER($X, F, \mathcal{M}, \epsilon, \delta, \alpha, \beta, n, p, p', p'', g, c$)

  **for** $t \leftarrow n, n+1, \dots$ **do**
    Let $\epsilon_t \leftarrow \frac{\sqrt{c}}{3\sqrt{\log(1/\delta)}} \frac{\epsilon}{t^{\frac{1}{2}+c}}$.
    Let $\beta_t \leftarrow \frac{\beta}{2t^2}$
    Let $\alpha_t \leftarrow g\left(\frac{1}{\epsilon_i t}\right)^p \log^{p''} n \log^{p'} \frac{1}{\beta}$
    Let $y_t \leftarrow \mathcal{M}(x_t, \epsilon_t, \alpha_t, \beta_t)$
    **for** $j \leftarrow 1, \dots, \ell_t$ **do**
      Output $y_t(f_{t,j})$
    **end for**
  **end for**

---

**Theorem 29.** Let $c > 0$ and let $\mathcal{M}$ be a $(p, p', p'', g)$-black box. Then for any database stream $X$ and stream of linear queries $F$, BBIMPROVER($X, F, \mathcal{M}, \epsilon, \delta, \beta, n, p, p', p'', g, c$) is $(\epsilon, \delta)$-differentially private for $\epsilon < 1$ and $(\{\alpha_t\}_{t \geq n}, \beta)$-accurate for sufficiently large constant $C$ and

$$\alpha_t \geq Cg\left(\frac{\log^{(p'/p)}(1/\beta)\sqrt{\log(1/\delta)}}{\sqrt{c}\epsilon t^{\frac{1}{2}-2c}}\right)^p.$$

*Proof.* We start with the privacy guarantee. BBIMPROVER runs $\mathcal{M}$ at each time $t$ with privacy parameters $\epsilon_t = \frac{\epsilon}{t^{\frac{1}{2}+c}}$. By Theorem 4, the total privacy loss is at most $\frac{1}{2}\sum_{t=n}^{\infty}\epsilon_t^2 + \sqrt{2\left(\sum_{t=n}^{\infty}\epsilon_t^2\right)\log(1/\delta)}$.

Note:

$$\sum_{t=n}^{\infty}\epsilon_t^2 = \frac{c\epsilon^2}{9\log(1/\delta)}\sum_{t=n}^{\infty}\frac{1}{t^{1+2c}} \leq \frac{c\epsilon^2}{9\log(1/\delta)}\frac{1}{cn^{2c}} \leq \frac{\epsilon^2}{9\log(1/\delta)}.$$

since $n \geq 1$ and $c > 0$. Then the privacy loss is at most:

$$\frac{1}{18\log(1/\delta)}\epsilon^2 + \frac{2\epsilon}{3\sqrt{\log(1/\delta)}}\sqrt{\log(1/\delta)} \leq \epsilon \tag{D.2}$$

since $\epsilon < 1$.

We next prove the accuracy of BBIMPROVER. For each time $t$, $\mathcal{M}(x_t, \epsilon_t, \alpha_t, \beta_t)$ is $\left(g\left(\frac{1}{\epsilon_t t}\right)^p \log^{p''} t \log^{p'} \frac{1}{\beta_t}, \beta_t\right)$-accurate. We simply plug in $\epsilon_t$ and $\beta_t$ to get our accuracy bound at time $t$:

$$\alpha_t = g\left(\frac{1}{\epsilon_t t}\right)^p \log^{p''} t \log^{p'} \frac{1}{\beta_t}$$

$$= g\left(\frac{3\sqrt{\log(1/\delta)}t^{\frac{1}{2}+c}}{\sqrt{c}\epsilon t}\right)^p \log^{p''} t \log^{p'} \frac{2t^2}{\beta}$$

$$\leq C_1 g\left(\frac{\sqrt{\log(1/\delta)}}{\sqrt{c}\epsilon t^{\frac{1}{2}-c}}\right)^p \log^{p''+p'} t \log^{p'} \frac{1}{\beta}$$

$$\leq C_2 g\left(\frac{\sqrt{\log(1/\delta)}}{\sqrt{c}\epsilon t^{\frac{1}{2}-2c}}\right)^p \log^{p'} \frac{1}{\beta},$$

for positive constants $C_1$ and $C_2$. The last line holds since $\log^{p''+p'} t = o(t^c)$ for any positive constants $c, p',$ and $p''$.

The accuracy of $\mathcal{M}(x_t, \epsilon_t, \beta_t)$ at time $t$ holds with probability $1 - \beta_t$. By a union bound, all accuracy guarantees will be satisfied with probability $1 - \sum_{t=n}^{\infty}\beta_t$, where,

$$\sum_{t=n}^{\infty}\beta_t = \sum_{t=n}^{\infty}\frac{\beta}{2t^2} \leq \frac{\pi^2}{12}\beta \leq \beta.$$

$\square$

# E ERM for Growing Databases

## E.1 ERM Background

Empirical risk minimization (ERM) is one of the most fundamental tasks in machine learning. In ERM, the task is to find a classifier from some set $\mathcal{C}$ that minimizes a loss function $\mathcal{L}$ on the sample data. Formally, we are given some data set $x_n = \{z_1, ..., z_n\} \in \mathcal{X}^n$, where each $z_i$ is sampled independently from some distribution $P$. We are also given some set $\mathcal{C}$ such that for $\theta \in \mathcal{C}$, the loss function $\mathcal{L}$ is defined as:

$$\mathcal{L}(\theta; x_n) = \frac{1}{n} \sum_{i=1}^{n} L(\theta; z_i)$$

where for all $z \in \mathcal{X}$, $L(\cdot; z)$ maps from $\mathcal{C}$ to $\mathbb{R}$. Common choices for $L$ include the $0 - 1$ loss, hinge loss, and the squared loss. We seek to find a $\hat{\theta}$ with small *excess empirical risk*, defined as

$$\hat{\mathcal{R}}_n(\hat{\theta}) = \mathcal{L}(\hat{\theta}; x_n) - \min_{\theta \in \mathcal{C}} \mathcal{L}(\theta; x_n) \tag{E.1}$$

In convex ERM, we assume that $L(\cdot; x)$ is convex for all $x \in \mathcal{X}$ and that $\mathcal{C}$ is a convex set. We will also assume that $\mathcal{X} \subseteq \mathbb{R}^p$. Convex ERM is convenient because finding a suitable $\hat{\theta}$ reduces to a convex optimization problem, for which there exist many fast algorithms. Some examples of ERM include finding a $d$-dimensional median and SVM.

ERM is useful due to its connections to the *true risk*, also known as the generalization error, which we define as:

$$\mathcal{R}(\theta) = \mathop{\mathrm{E}}_{x \sim P} \left[ L(\theta; x) \right]$$

That is, the loss function will be low in expectation on a new data point sampled from $P$. We can also define the *excess risk* of a classifier $\hat{\theta}$:

$$\mathrm{ExcessRisk}(\hat{\theta}) = \mathop{\mathrm{E}}_{x \sim P} \left[ L(\hat{\theta}; x) \right] - \min_{\theta \in \mathcal{C}} \mathop{\mathrm{E}}_{x \sim P} \left[ L(\theta; x) \right]$$

ERM finds classifiers with low excess empirical risk, which in turn often have low excess risk. The following theorem relates the two. For completeness, we first give some definitions relating to convex empirical risk minimization sections. A convex body $\mathcal{C}$ is a set such that for all $x, y \in \mathcal{C}$ and all $\lambda \in [0, 1]$, $\lambda x + (1 - \lambda)y \in \mathcal{C}$. A vector $v$ is a subgradient of a function $L$ at $x_0$ if for all $x \in \mathcal{C}$, $L(x) - L(x_0) \geq \langle v, x - x_0 \rangle$. A function $L : \mathcal{C} \to \mathbb{R}$ is $G$-Lipschitz if for all pairs $x, y \in \mathcal{C}$, $|L(x) - L(y)| \leq G \|x - y\|_2$ $L$ is $\Delta$-strongly convex on $\mathcal{C}$ if for all $x \in \mathcal{C}$ and all subgradients $z$ at $x$ and all $y \in \mathcal{C}$, we have $L(y) \geq L(x) + \langle z, y - x \rangle + \frac{\Delta}{2} \|y - x\|_2^2$. $L$ is $B$-smooth on $\mathcal{C}$ if for all $x \in \mathcal{C}$, for all subgradients $z$ at $x$ and for all $y \in \mathcal{C}$, we have $L(y) \leq L(x) + \langle z, y - x \rangle + \frac{B}{2} \|y - x\|_2^2$. We denote the diameter of a convex set $\mathcal{C}$ by $\|\mathcal{C}\|_2 = \arg\max_{x,y \in \mathcal{C}} \|x - y\|_2$.

**Theorem 30 ([SSSSS09]).** For $G$-Lipschitz and $\Delta$-strongly convex loss functions, with probability at least $1 - \gamma$ over the randomness of sampling the data set $X_n$, the following holds:

$$\mathrm{ExcessRisk}(\hat{\theta}) \leq \sqrt{\frac{2G^2}{\Delta} \hat{\mathcal{R}}_n(\hat{\theta})} + \frac{4G^2}{\gamma \Delta n}$$

Moreover, we can generalize this result to any convex and Lipschitz loss function $L$ by defining a regularized version of $L$, called $\tilde{L}$, such that $\tilde{L}(\theta; x) = L(\theta; x) + \frac{\Delta}{2} \|\theta\|_2^2$. Then $\tilde{L}$ is $(L + \|\mathcal{C}\|_2)$-Lipschitz and $\Delta$-strongly convex. Also note that:

$$\mathrm{ExcessRisk}_L(\theta) \leq \mathrm{ExcessRisk}_{\tilde{L}}(\theta) + \frac{\Delta}{2} \|\mathcal{C}\|_2^2$$

Thus, ERM finds classifiers with low true risk in these settings.

## E.2 BBIMPROVER for ERM

For ERM in the dynamic setting, we want a classifier $y_t$ at every time $t \geq n$ that achieves low empirical risk on the current database, and we want the empirical risk of our classifiers to improve over time, as in the static case. Note that the dynamic variant of the problem is strictly harder because we must produce classifiers at every time step, rather than waiting for sufficiently many new samples to arrive. Releasing classifiers at every time step degrades privacy, and thus requires more noise to be added to preserve the same overall privacy guarantee. Nonetheless, we will compare our private growing algorithm, which provides accuracy bounds for every time step from $n$ to infinity simultaneously, to private static algorithms, which are only run once.

In ERMG, our algorithm for ERM in the dynamic setting, the sole query of interest is the loss function $\mathcal{L}$ evaluated on the current database. At each time $t$, ERMG receives a single query $f_t$, where $f_t$ evaluated on the database is $x_t(f_t) = \min_{\theta \in \mathcal{C}} \mathcal{L}(\theta; x_t)$. The black box outputs $y_t$, which is a classifier from $\mathcal{C}$ that can be used to evaluate the single query $y_t(f_t) = \mathcal{L}(y_t; x_t)$. Our accuracy guarantee at time $t$ is the difference between $y_t(f_t)$ and $x_t(f_t)$:

$$\alpha_t = \mathcal{L}(y_t; x_t) - \min_{z \in \mathcal{C}} \mathcal{L}(z; x_t).$$

This expression is identical to the excess empirical risk $\hat{\mathcal{R}}_t(y_t)$ defined in Equation (E.1). Thus accurate answers to queries are equivalent to minimizing empirical risk. Our accuracy bounds are stated in Theorem 31. The results come from instantiating (the more general) Theorem 29. The differing assumptions on $\mathcal{L}$ allow us to use different $(p, g)$-black boxes with different input parameters in each case. We use the static $(\epsilon, 0)$-DP algorithms of [BST14] as black boxes. We compare our growing bounds to these static bounds in Table 3.[5] Since ERMG provides $(\epsilon, \delta)$-differential privacy, we also include static $(\epsilon, \delta)$-DP bounds from [BST14, KST+12] for comparison in Table 3. The static bounds are optimal in $d, t$, and $\epsilon$ up to log factors.

**Theorem 31.** Let $\mathcal{L}$ be a convex loss function that is 1-Lipschitz over some set $\mathcal{C}$ with $||\mathcal{C}||_2 = 1$. Then for any stream of databases $X$ with points in $\mathbb{R}^d$, ERMG$(X, \mathcal{L}, \mathcal{C}, \epsilon, \delta, \beta, n)$ is $(\epsilon, \delta)$-differentially private and with probability at least $1 - \beta$ produces classifiers $y_t$ for all $t \geq n$ that have excess empirical risk bounded by:

$$\hat{\mathcal{R}}_t(y_t) \leq \frac{d\sqrt{\log(1/\delta)} \log \frac{1}{\beta}}{\sqrt{c}\epsilon t^{\frac{1}{2}-c}}$$

If $\mathcal{L}$ is also $\Delta$-strongly convex,

$$\hat{\mathcal{R}}_t(y_t) \leq \frac{d^2 \log(1/\delta) \log^2 \frac{1}{\beta}}{\sqrt{c}\Delta\epsilon^2 t^{1-c}}$$

where $c$ is any positive constant.

*Proof.* When the loss function is 1-Lipschitz and $||\mathcal{C}||_2 = 1$, the $\epsilon$-differentially private static algorithm from [BST14] is a $(p, g)$-black box for $p = 1$ and $g = d$. When the loss function is also $\Delta$-strongly convex, the $\epsilon$-differentially private static algorithm from [BST14] is a $(p, p', p'', g)$-black box for $p = 2$, $p' = 2$, $p'' = 1$, and $g = d^2/\Delta$. The bounds of Theorem 31 come from instantiating Theorem 29 using each of these black boxes. $\square$

Note that the bounds we get for the growing setting have the same dependence on $\epsilon, \beta$, and $\Delta$ and better dependence on $\delta$. The dependence on $t$ in our bound is roughly the square root of that in the static bounds. Compared to the static $(\epsilon, 0)$-DP bounds, our dependence on $d$ is the same, while the dependence is squared relative to the static $(\epsilon, \delta)$-DP bounds.

Given that the growing setting is strictly harder than the static setting, it is somewhat surprising that we have no loss in most of the parameters, and only minimal loss in the size of the database $t$. Thus, for ERM, performance in the static setting largely carries over to the growing setting.

Table 3: Comparison of excess empirical risk upper bounds in the static case ([BST14]) versus the dynamic case (this work) for a database of size $t$ for different assumptions on the loss function $L$. (Database entries are sampled from $\mathbb{R}^d$, and $c$ is any positive constant. We ignore leading multiplicative constants and factors of $\log\log\frac{1}{\beta}$ in the static bounds. As in [BST14], we assume $\delta < 1/n$ for simplicity.)

| Assumptions | Static $(\epsilon, 0)$-DP | Static $(\epsilon, \delta)$-DP | Dynamic $(\epsilon, \delta)$-DP (our results) |
|---|---|---|---|
| 1-Lipschitz and $\|\mathcal{C}\| = 1$ | $\dfrac{d\log\frac{1}{\beta}}{\epsilon t}$ | $\dfrac{\sqrt{d}\log^2(t/\delta)\log\frac{1}{\beta}}{\epsilon t}$ | $\dfrac{d\sqrt{\log(1/\delta)}\log\frac{1}{\beta}}{\sqrt{c}\epsilon t^{\frac{1}{2}-c}}$ |
| ... and $\Delta$-strongly convex (implies $\Delta \leq 2$) | $\dfrac{d^2(\log t)\log^2\frac{1}{\beta}}{\Delta\epsilon^2 t^2}$ | $\dfrac{d\log^3(t/\delta)\log^2\frac{1}{\beta}}{\Delta\epsilon^2 t^2}$ | $\dfrac{d^2\log(1/\delta)\log^2\frac{1}{\beta}}{\sqrt{c}\Delta\epsilon^2 t^{1-c}}$ |

## Footnotes

[4]Proposition 10 from the extended version http://www.maths.lancs.ac.uk/jameson/gammainc.pdf

[5]To get the static bounds, we use Appendix D of [BST14], which converts bounds on expected excess empirical risk to high probability bounds.