[Reviews · NeurIPS 2018]

Reviewer 1



This paper studies differential privacy for dynamic databases, which have elements added over time. I think this is an important direction, as this is perhaps a more realistic model than the static setting, when it comes to practical deployment. While there was some prior work for very basic dynamic settings, this paper makes a much more significant contribution in this direction. The authors provide a modification of Private Multiplicative Weights for growing databases. Beyond this, they provide black-box methods for converting static mechanisms to dynamic mechanisms. They also develop a number of other primitives to the dynamic setting (including Numeric Sparse and Above Threshold of [? ] (broken citation in line 227...)). In short, I think this makes a number of important contributions to an area of practical interest, and recommend acceptance.

Reviewer 2



This paper studies how to answer queries on a dynamically growing database while satisfying differential privacy. It first offers a concrete algorithm, along with formal privacy and utility guarantees, for the linear query case. This algorithm, PMWG, relies heavily upon PMW, which maintains a public histogram to answer a number of linear queries exponential in the database size. PMWG then tweaks PMW to essentially renormalize (independent of the new data) this histogram upon the addition of new data as if that data is uniform -- thereby getting around the histogram's "overfitting" to old data -- and loses only constant factors in accuracy. Second, the paper provides a general black-box transformation from any sufficiently private and accurate static algorithm to a private and accurate dynamic one, presenting ERM as an application. The paper actually provides two transformations with slightly different accuracy guarantee, but they share a similar structure: use a sequence of epsilons {eps_i} that decays as the database size increases, so that the epsilons sum to a fixed epsilon while the database growth allows a constant (or increasing) level of accuracy even with ever-tighter privacy requirements. Along the way, the paper introduces and uses a number of dynamic versions of canonical static differentially private algorithms, including private multiplicative weights and sparse vector. I did not check the proofs in the Supplement, but I did look at all Lemma and Theorem statements. Positive: The dynamic database setting is a natural and practical one for differential privacy, and I think this paper is a good first cut on the topic. The actual theoretical results are not especially surprising -- I think most experts in the area already have some hazy idea that algorithms like PMWG, BBSchedule, and BBImprove exist. Still, it's good to know that the vague intuitive approach actually works out formally, and it's not like the required analysis is trivial. I think the fact that this paper actually works it out formally for a few dynamic versions of static workhorse differentially private algorithms makes it a good contribution to the field. In my opinion, PMWG and the dynamic versions of sparse-* algorithms are enough to be a fine NIPS paper. Negative (in order of decreasing importance): 1. Section 4, on BBScheduler and BBImprove, are hard for me to follow. Some of this is perhaps intrinsic to the problems, but for example I still don't know what that g really means, and the difference between BBScheduler and BBImprove is kind of muddy -- honestly, I think it would make more sense to forget BBScheduler altogether, though I don't expect this to happen. 2. Why do the log(n) factors in the dynamic accuracy bounds for PMWG (Theorem 5) not appear in Theorem 1, whose surrounding text repeatedly describes this error as "tight" wrt PMW? I don't think this factor is a big deal, but I'm confused as to why it apparently doesn't count. 3. The paper's exposition has room for improvement. As mentioned above, most of these algorithms have reasonably intuitive interpretations, but these weren't very clear on the first read, e.g. just reframing "updated as if the new data came from a uniform distribution" as a version of normalization or saying "epsilon decay is tied to database growth so that the two balance out/database growth eventually dominates" would have helped me. These are the punchlines, no? It's nice to have memorable takeaways, as is it's easy to miss them and get lost in all of the parameters floating around. 4. The intro is redundant with the abstract and at times is almost a copy. To some extent this is an issue in every paper, but given the (IMO) better uses of space mentioned above, perhaps some reorganization would help. Section 4.2.1, which quickly discusses private ERM for growing databases, feels pretty extraneous -- everybody reading this paper knows what ERM is, no? Seems like this can be a single sentence ("We include details for an example application of BBImprove for private ERM on growing databases in the Supplement.") 5. The organization of the supplement reverses the order of discussion from the main body -- the main body covers PMWG first and then the black-box transformation, and the Supplement does the opposite. Minor, but odd. 6. There are several broken references, e.g. on lines 108, 227, 545. The main body also appears to be too long (???) stretching onto a 9th page. Typos "We give the parameters the" on line 292, "Accurary" on line 588. Overall: Though I've provided more negative points than positive ones, the positives outweigh the negatives (which are mostly presentation-focused) for me. This paper provides a nice (if not very surprising) analysis of a useful problem, and develops the (scant) existing theory on "dynamic database" differential privacy along several fronts. I therefore vote to accept. ---Update After Response--- I read the authors' response and found the clarifications useful. I'm keeping my score as is, and am willing to argue some for acceptance.

Reviewer 3



This paper explores differential privacy (DP) under the dynamical scenario where the datasets grow in time. Differential privacy literature mostly considers the static case so this is a fairly unexplored topic. The authors propose a differentially private multiplicative weights method, i.e., an adaptive stream of linear queries, for the dynamical setting. They also provide transformations for private static algorithms (both pure and approximate DP) to work in the dynamical case and study their accuracy. It is a very well-written paper and good contribution, filling an important gap in the literature.